

# Distributions of bacteriohopanepolyols in lakes and coastal lagoons of the Azores Archipelago

Nora Richter[1,2*], Ellen C. Hopmans[1], Danica Mitrović[1], Pedro M. Raposeiro[3,4], Vítor Gonçalves[3,4], Ana C. Costa[3,4], Linda A. Amaral-Zettler[1,2,5], Laura Villanueva[1,6], and Darci Rush[1]

[1]Department of Marine Microbiology and Biogeochemistry, NIOZ Royal Netherlands Institute for Sea Research, 1790 AB Den Burg, The Netherlands

[2]Department of Earth, Environmental and Planetary Sciences, Brown University, Providence, USA

[3]Centro de Investigação em Biodiversidade e Recursos Genéticos, CIBIO, InBIO Laboratório Associado, BIOPOLIS Program in Genomics, Biodiversity and Land Planning – UNESCO Chair – Land Within Sea: Biodiversity & Sustainability in Atlantic Islands, Pólo dos Açores, Portugal

[4]Faculdade de Ciências e Tecnologia da Universidade dos Açores, Ponta Delgada, Açores, Portugal

[5]Department of Freshwater and Marine Ecology, Institute for Biodiversity and Ecosystem Dynamics, University of Amsterdam, Amsterdam, The Netherlands

[6]Department of Earth Sciences, Utrecht University, Utrecht, The Netherlands

*Correspondence to*: Nora Richter (nora.richter@nioz.nl)

**Abstract.** Bacteriohopanepolyols (BHPs) are a diverse class of lipids produced by bacteria across a wide range of environments. In this study, we aim to further identify BHPs related to ecological niches and/or specific bacteria by characterizing the distribution of BHPs in suspended particulate matter (SPM) of the water column and in sediments in a range of lakes and coastal lagoons from the Azores Archipelago, as well as in a co-culture enriched for methanotrophs. Sediment samples from Azorean lakes with low oxygen conditions during the summer months (i.e., Azul, Verde, Funda, and Negra) contain relatively high abundances of BHPs that are typically associated with methane-oxidizing (methanotrophic) bacteria (i.e., aminotetrol, aminopentol, and methylcarbamate-aminopentol), as well as the ethenolamine-BHPs (i.e., ethenolamine-BHpentol and ethenolamine-BHhexol) and the formylated aminoBHPs. Both ethenolamine-BHPs and formylated aminoBHPs were also detected in a co-culture enriched for methanotrophs obtained from a lake. In the SPM of all water columns, bacteriohopanetetrol (BHT), BHT-cyclitol ether, and aminotriol are the dominant BHPs. In SPM from Lake Funda, nucleoside BHPs (i.e., Me-adenosylhopane$_{HG-diMe}$, N1-methylinosylhopane, 2Me-N1-inosylhopane, and Me-N1-inosylhopane) are present in low abundance or absent under oxic conditions but increase in concentration near the chemocline, suggesting potential *in situ* production of these nucleoside BHPs rather than an allochthonous origin. In contrast, sediments from shallow, well-mixed lakes (i.e., Empadadas, São Jorge, and Lomba) contain higher abundances of adenosylhopane and N1-methylinosylhopane, which likely originate from bacteria living in nearby soils. In the coastal lagoons, Cubres East and West, methoxylated-BHTs were detected, and higher abundances of ethenolamine-BHT were observed. This study highlights the diversity of BHPs in lakes and coastal lagoons and their potential as taxonomic markers for bacteria associated with certain ecological niches, which can be preserved in sedimentary records.



## 1 Introduction

Bacteriohopanepolyols (BHPs) are pentacyclic triterpenoids found in the cell membrane of many gram-negative and gram-positive bacteria (Rohmer et al., 1984). Hopanoid derivatives of BHPs are considered to be some of the most abundant lipids on Earth and in the geologic record (Ourisson and Albrecht, 1992). BHPs are involved in various cell physiological processes (Welander et al., 2009; Sáenz, 2010; Doughty et al., 2011; Sáenz et al., 2012;

Welander and Summons, 2012) and, due to their structural diversity, show potential as chemotaxonomic markers (Kusch and Rush, 2022). Functionalized BHPs were identified in sediment records that span 1.2 Ma (Talbot et al., 2014) as well as in Eocene Cobham Lignite (ca. 56 Ma; Talbot et al., 2016), highlighting their preservation potential as biomarkers in sedimentary records.

Recent studies and advancements in BHP analysis highlight the wide distribution and diversity of BHPs in both marine and terrestrial environments (Talbot and Farrimond, 2007; Pearson et al., 2009; Sáenz et al., 2011; Kusch et al., 2019; Hopmans et al., 2021). Bacteriohopanetetrol (BHT), BHT-cyclitol ether (CE), and aminotriol, for instance, are ubiquitous in environmental samples and culture studies, and are not associated with any specific organisms or environments (e.g., Talbot et al., 2003; Talbot and Farrimond, 2007; Zhu et al., 2011). Nucleoside

BHPs, (formerly known as adenosylhopanes) are typically associated with soils, and are used to trace soil inputs to riverine and marine environments using a ratio of nucleosides to the more ubiquitous BHT, known as the $R_{soil}$ index (Taylor and Harvey, 2011; Zhu et al., 2011; De Jonge et al., 2016; Kusch et al., 2019). Recent studies, however, show that certain nucleoside BHPs can also be produced in the marine environment under low-oxygen conditions, complicating interpretations of the $R_{soil}$ index (Kusch et al., 2021b). Hopanoids methylated at the C-2

position are degradation products of 2-methylated-BHPs (e.g., 2Me-BHT), and are often considered diagnostic for cyanobacteria in the geologic record (Summons et al., 1999). 2Me-BHPs are produced by cyanobacteria (Talbot et al., 2008), however, the gene responsible for BHP methylation at the C-2 position was identified in a wide-range of other bacterial phyla (Welander et al., 2010), suggesting multiple biological sources. This highlights the complexity of BHPs in the environment, and the need for more studies to evaluate the potential of BHPs as

biomarkers for specific bacterial groups.

Aerobic methane-oxidizing bacteria (MOB) regulate methane emissions in seasonally stratified lakes (Bastviken et al., 2008; Oswald et al., 2015; Guggenheim et al., 2020) and meromictic lakes (Oswald et al., 2016) by oxidizing methane before it reaches the atmosphere. MOB in lacustrine environments typically fall within the two

phylogenetic groups in the phylum *Proteobacteria*: Type I (members of *Gammaproteobacteria*) and Type II (members of *Alphaproteobacteria*; Hanson and Hanson, 1996). They are known to produce a wide-range of potentially diagnostic BHPs (Rohmer et al., 1984), including aminotetrol and aminopentol (Neunlist and Rohmer, 1985b, c; Cvejic et al., 2000; Talbot et al., 2001; van Winden et al., 2012), the recently identified methylcarbamate-aminoBHPs (Rush et al., 2016), and related 3β-methylated-BHPs (Neunlist and Rohmer, 1985b;

Zundel and Rohmer, 1985; Cvejic et al., 2000). However, minor amounts of these BHPs are also produced by sulfur-reducing bacteria and nitrite-oxidizing bacteria (Blumenberg et al., 2006; Elling et al., 2022). Few studies have evaluated the BHP composition of lacustrine methanotrophs nor the utility of these biomarkers for tracing MOB in lakes.



In addition to their potential as taxonomic markers, BHP relative abundance was observed to change under certain environmental conditions. For instance, microcosm and mesocosm experiments that enriched for methanotrophs demonstrate that increasing temperatures led to increased concentrations of aminotriol, aminotetrol, and aminopentol (Osborne et al., 2017; van Winden et al., 2020), whereas a decrease in the relative abundance of unsaturated hopanoids was observed in culture experiments (Bale et al., 2019). Shifts in BHP compositions were

also used to interpret changes in salinity (Coolen et al., 2008), redox conditions (Blumenberg et al., 2013; Matys et al., 2017; Rush et al., 2019; Zindorf et al., 2020), and soil inputs (Blumenberg et al., 2013) in sediment records. BHPs and BHP-producers are particularly diverse in lakes (Farrimond et al., 2000; Watson and Farrimond, 2000; Talbot et al., 2003; Talbot and Farrimond, 2007; O'Beirne et al., 2022), yet few studies have associated BHPs with specific lacustrine settings.


   The purpose of this study is to characterize the distribution of BHPs in lakes and coastal lagoons with the aims of identifying BHPs that are specific to lacustrine bacteria, with a focus on MOB, and/or specific ecological niches, and of testing whether nucleoside BHPs are produced *in situ* in lakes. We also analyzed BHPs from a methanotrophic-methylotrophic co-culture enriched from a lake water column to identify potential novel BHPs

related to lacustrine methanotrophy. We also collected and analyzed sediment and water column suspended particulate matter (SPM) from eight lakes and two coastal lagoons on three different islands in the Azores Archipelago. The Azores Archipelago consists of nine islands with 88 lakes that occur in either topographically depressed areas or volcanic depressions, and range from severely impacted by humans to relatively "pristine" (Pereira et al., 2014). The diversity in lake-types and comparison with marine-influenced sites, i.e. the coastal

lagoons, in this study provide an ideal setting to investigate the partitioning of BHPs in the environment and their potential as biomarkers.

## 2 Materials & Methods

### 2.1 Sample Collection

   Samples were collected in lakes and lagoons on three different islands in the Azores Archipelago in June 2018

(Fig. 1 & Table 1): São Miguel (lakes Azul, Verde, and Empadadas), São Jorge (lake São Jorge and lagoons Cubres East and Cubres West), and Flores (lakes Funda, Lomba, and Negra). June is considered part of the dry season in the Azores, which ranges from April to August. From September to March, the Azores Archipelago receives more rain and is exposed to high winds (Santos et al., 2004; Hernández et al., 2016). During the summer months, deeper lakes such as Azul, Verde, Funda, and Negra stratify and the bottom water becomes suboxic and,

in some years, even fully anoxic (Gonçalves, 2008; Gonçalves et al., 2018). In contrast, shallow lakes are typically considered polymictic but might undergo short periods of stagnation during the summer months

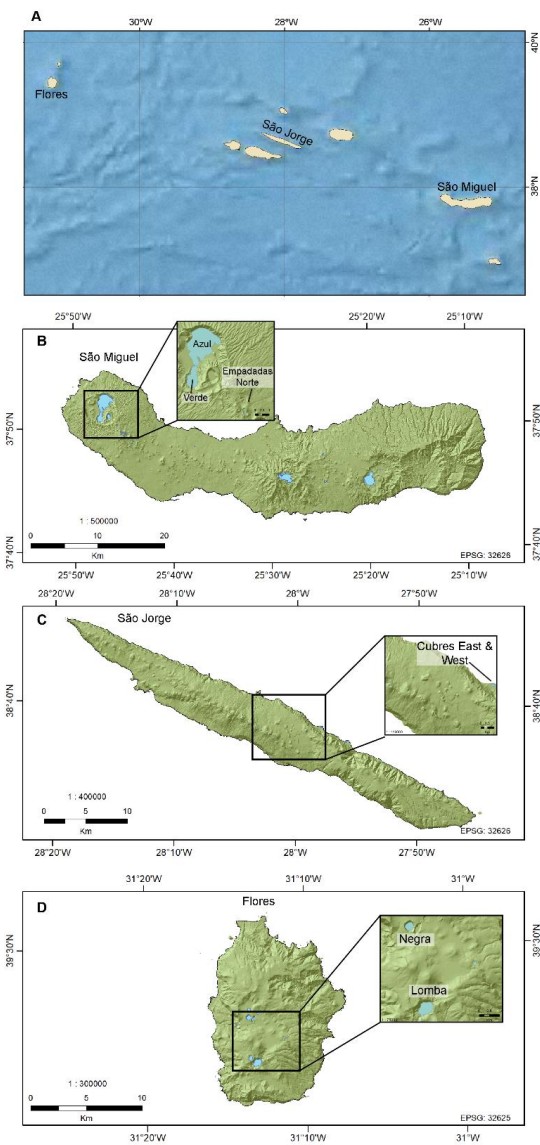

**Figure 1: (A) The Azores Archipelago with the location of lakes and coastal sites discussed in this study. Our study sites include (B) lakes Azul, Verde, and Empadadas Norte on São Miguel Island, (C) São Jorge lake and the coastal lagoons (i.e., Cubres East and Cubres West) on São Jorge Island, and (D) lakes Negra, Lomba, and Funda on Flores Island (maps generated in ArcGIS 10.3).**





**Table 1: Location and characteristics of lakes and lagoons discussed in this study. The annual average water column properties are based on long-term monitoring data from 2003-2017 (University of the Azores monitoring program) where TP = total phosphorous and TN = total nitrogen.**

| | | Site information | | | | | | | Annual Average Water Column properties | | | | |
|---|---|---|---|---|---|---|---|---|---|---|---|---|---|
| Island | Lake/Lagoon | Lat. (°N) | Lon. (°W) | Alt. (masl) | Max. Depth (m) | Trophic State* | Temp. (°C) | pH | TP (µg/L) | TN (µg/L) | NH$_4$ (µg/L) | NO$_3$ (µg/L) | NO$_2$ (µg/L) |
| São Miguel | Azul | 37.87 | -25.78 | 260 | 25.4 | Mesotrophic | 16.55 | 7.51 | 23.13 | 0.43 | 86.83 | 0.46 | 16.34 |
| São Miguel | Verde | 37.84 | -25.79 | 262 | 23.5 | Eutrophic | 15.56 | 7.74 | 44.96 | 0.59 | 263.94 | 0.45 | 24.26 |
| Sao Miguel | Empadadas Norte | 37.82 | -25.75 | 762 | 3.3 | Eutrophic | 14.3 | 6.90 | 22.67 | 0.45 | 15.19 | 0.75 | 20.20 |
| São Jorge | São Jorge | 38.65 | -28.07 | 900 | 2.5 | - | - | - | - | - | - | - | - |
| São Jorge | Cubres East | 38.64 | -27.97 | 3 | 2.0 | - | - | - | - | - | - | - | - |
| São Jorge | Cubres West | 38.64 | -27.97 | 3 | 2.0 | - | - | - | - | - | - | - | - |
| Flores | Funda | 39.41 | -31.22 | 364 | 31.9 | Eutrophic | 14.69 | 7.72 | 41.84 | 0.62 | 248.70 | 0.69 | 39.89 |
| Flores | Lomba | 39.43 | -31.19 | 651 | 15.3 | Mesotrophic | 14.63 | 7.01 | 24.16 | 0.38 | 19.57 | 0.38 | 22.20 |
| Flores | Negra | 39.44 | -31.23 | 540 | 115.0 | Eutrophic | 14.66 | 7.99 | 98.98 | 0.60 | 86.40 | 0.80 | 24.83 |

*University of the Azores monitoring program (2003-2017); Cordeiro et al. (2020)

(Gonçalves, 2008). The lakes sampled for this study are either mesotrophic or eutrophic and range in water column depth from 2 to 115 m. During the summer months, lakes Verde and Funda experience large cyanobacterial blooms (Cordeiro et al., 2020).


Suspended particulate matter was collected from the surface water, near the oxycline/chemocline at the time of sampling, and from the bottom water of the lake without disturbing the sediment-water interface. Water (5 to 10 L) was immediately filtered through Whatman GF/F 142 mm 0.7µm pore size filters and frozen in liquid nitrogen. Surface sediment samples (0-1 cm) were also obtained at the same time using a gravity corer (UWITEC-90mm,

Austria). In Azul and Verde, surface sediments were collected near the lake shore and in the deepest part of the lake. Replicate samples were collected for sediment cores from lakes Azul (n = 2), Empadadas (n = 4) and Negra (n = 2) and analyzed for reproducibility. All samples were immediately frozen in a liquid nitrogen dewar and transported to the Royal NIOZ Netherlands Institute for Sea Research, where SPM and sediment samples were stored at -80°C and -40°C, respectively, until analysis. At the time of sample collection, water column profiles

were obtained using a Hydrolab (Horiba-U50) for dissolved oxygen content (percent saturation and mg/L), temperature, salinity, conductivity, pH, and turbidity (NTU). Additional long-term data was obtained from the Universidad dos Açores long-term monitoring program (data from 2003-2020).

### 2.2 Cultivation of Methanotroph-Methylotroph co-culture

An enrichment co-culture consisting of a methanotroph (43% *Methylobacter* sp.) and a methylotroph (21%

*Methylotenera* sp.) was previously isolated from a seasonally stratified and hypereutrophic lake (Lacamas Lake, WA, USA; van Grinsven et al., 2020). This culture was grown in triplicate under oxic conditions in the dark at 15 °C on a nitrate mineral salts (NMS) medium (Whittenbury et al., 1970). For lipid extractions, the cultures were grown on 250 mL NMS media in 580 mL acid-washed and autoclaved glass pressure bottles with butyl rubber stoppers with an addition of methane (16 mL, 99.99% pure) to the headspace. After the methane gas was





consumed, the cultures were filtered onto muffled Whatman GF/F 47 mm 0.3 µm pore size filters and frozen at -
       80 °C. The methylotroph identified in the co-culture is closely related to *Methylotenera versatilis* and
       *Methylotenera mobilis* (van Grinsven et al., 2020). Therefore, we obtained freeze-dried biomass of *Methylotenera*
       *mobilis* (DSM 17540) from the Deutsche Sammlung von Mikroorganismen und Zellkulturen (DSMZ) culture
       collection to investigate the presence of BHPs in this methylotroph.

**2.3 Lipid extraction**

       All samples were freeze-dried and extracted using a modified Bligh-Dyer method (Bligh and Dyer, 1959; Bale et
       al., 2021). Samples were submerged in methanol (MeOH), dicholoromethane (DCM), and phosphate buffer
       (2:1:0.8, v:v:v) and ultrasonically extracted twice. The solvent was collected in a separate flask after each
       extraction. DCM and phosphate buffer were added to the resulting solvent to obtain a new volume ratio of 1:1:0.9

(v:v:v). The DCM layer was collected and the aqueous layer was washed two more times with DCM. The
       extraction was repeated using MeOH:DCM:aqueous trichloroacetic acid solution (2:1:0.8, v:v:v) following the
       same procedure described above. The combined DCM layers were dried under $N_2$ gas and stored at -20 °C until
       analysis. Prior to analysis, deuterated diacylglyceryltrimethylhomoserine (DGTS D-9; Avanti® Polar Lipids,
       USA) was added to the extracts as an internal standard and the samples were re-dissolved in MeOH:DCM (9:1,

v:v) before filtering the samples through a 0.45 µm regenerated cellulose syringe filter (4 mm diameter; Grace
       Alltech, Deerfield, IL).

       **2.4 UHPLC/HRMS analysis**

       Samples were analyzed following the methods described in Hopmans et al. (2021). Briefly, samples were analyzed
       on an Agilent 1290 Infinity I UHPLC coupled to a quadrupole-orbitrap HRMS equipped with an Ion Max source

and heated ESI probe (HESI) (ThermoFisher Scientific, Waltham, MA). Separation was achieved on an Acquity
       C18 BEH column (2.1 x 150 mm, 1.7 µm particle; Waters) and pre-column. The solvent system consisted of (A)
       MeOH:$H_2O$ (85:15) and (B) MeOH:isopropanol (1:1) with both containing 0.12 % (v/v) formic acid and 0.04 %
       (v/v) aqueous ammonia. Lipids were detected using positive ion monitoring of *m/z* 350-2000 (resolution 70,000
       ppm at *m/z* 200) with an inclusion list of calculated exact masses of BHPs: 171 for sediment samples (one Lake

Verde sediment extract was re-run with 421 exact masses for confirmation) and 357 for water column samples.
       For untargeted BHP detection and identification, we used data dependent MS[2] (isolation window 1 *m/z*; resolution
       17,500 ppm at *m/z* 200) of the 10 most abundant ions for a total cycle of ca. 1.2 s and dynamic exclusion (6 s)
       with a 3 ppm mass tolerance. To obtain optimal fragmentation of BHPs, we used a stepped normalized collision
       energy of 22.5 and 40. Mass calibration was performed every 48 h using a Thermo Scientific Pierce LTQ Velos

ESI Positive Ion Calibration Solution.

       BHPs were identified based on their retention time, exact mass and fragmentation spectra. Integrations were
       performed on (summed) mass chromatograms (within 3 ppm mass accuracy) of relevant molecular ions ([M+H]$^+$,
       [M+NH$_4$]$^+$, and [M+Na]$^+$). We used an internal standard for normalization between sample runs; however, we do

not have an internal standard for BHP quantification, therefore all BHPs are reported as response units (RU).
       BHPs identified in water column and surface sediment samples are normalized to liters of water filtered (L) and
       grams of freeze-dried sediment (g), respectively.





### 2.5 Elemental analysis

Total organic carbon (TOC) and total nitrogen (TN) content was measured for the surface sediments (n=18) as
described in Mitrović et al. (*in review*). Briefly, the samples were decalcified, powdered, and freeze-dried prior to
analysis. All samples were analyzed using an Elementar Vario Isotope Cube directly linked to Isotope Ratio Mass
Spectrometer (IRMS) Elementar Isoprime vision. Samples for total organic carbon (TOC) were pretreated with
excess 2M hydrochloric acid (HCl) to remove carbonates and were put on a shaker overnight. The resulting
samples were neutralized with bidistilled water and freeze-dried. TOC was measured in duplicate with a
reproducibility of <0.5 %.

### 2.6 $R_{soil}$ Index

A soil index ($R_{soil}$) was developed to trace inputs of terrestrial-derived BHPs into the marine environment (Zhu et
al., 2011). This index relies on the observations that BHT is present in higher abundance relative to nucleosides
in the marine environment than in the soils (Pearson et al., 2009; Rethemeyer et al., 2010; Zhu et al., 2011). The
index is calculated as follows:

$$R_{soil} = \frac{\text{(soil-marker BHPs)}}{\text{(soil-marker BHPs+BHT)}} . \tag{1}$$

Previously, "soil-marker BHPs" included only adenosylhopane, N1-methylinosylhopane (i.e., type-2
adenosylhopane after Hopmans et al., 2021), adenosylhopane$_{HG-Me}$ (i.e., type-3 adenosylhopane after Hopmans et
al., 2021) and their ring-methylated counterparts. BHT in the $R_{soil}$ equation only refers to the isomer with 17,
21(H), 22$R$, 32$R$, 33$R$, 34$S$ stereochemistry. Previous high latitude studies have revised this index, $R'_{soil}$, to exclude
methylated nucleosides (Doğrul Selver et al., 2012):

$$R'_{soil} = \frac{\text{(adenosylhopane + N1-methylinosylhopane + adenosylhopane}_{HG-Me})}{\text{(adenosylhopane + N1-methylinosylhopane + adenosylhopane}_{HG-Me}\text{+BHT)}} . \tag{2}$$

In this study, we tested both of these indices at our sample sites.

### 2.7 Data visualization and statistical analyses

Non-metric multidimensional scaling (NMDS) analysis was conducted using a Bray-Curtis dissimilarity matrix
to visualize the similarities between BHP compositions at our sample sites. All analyses were performed and
visualized in R (version R-4.2.1; R Core Team, 2022) using the vegan package (version 2.5-7; Oksanen et al.,
2020) and ggplot2 (version 3.3.5; Wickham, 2016). NMDS plots were generated for both the water column and
sediment samples. To test whether there was a significant difference between our sample sites and sample site-
types (for sediment samples: deep lakes, shallow lakes, and coastal environments; for water column: surface water,
chemocline, and bottom water), we performed analysis of similarities (ANOSIM) tests in R using the vegan
package for the sediment samples and water column samples. A Bray-Curtis dissimilarity matrix was used for
these tests.



## 3 Results & Discussion

### 3.1 Diversity of BHPs in lake surface sediments & water columns

At our study sites, we identified 83 BHPs in total (Table A1). For the identification of novel BHPs, we discuss the carbon positions, rings, and modifications to the side-chain of the core BHP structure as shown in Fig. B1. The novel BHPs are briefly discussed here and described in more detail in Appendix B. In the Azorean lakes, BHPs were more diverse in surface sediments (average 51 BHPs/sediment sample) than in the water column SPM (average 11 BHPs/SPM; Fig. 2).


Nucleoside BHPs were particularly diverse in the water column SPM and surface sediments in lakes Empadadas, São Jorge, and Lomba. We identified fourteen nucleoside BHPs, including three inosylhopanes, described by Hopmans et al. (2021; Appendix B, Fig. B2 & B3). Further, we identified five additional adenosyl-BHP isomers (Fig. B2), including early eluting isomers (peaks b*, f*, g*, and k*). We also detected four additional inosylhopane

isomers in surface sediments from Lake Empadadas with modifications in the position of the methyl group on the ring structure and/or head group (Fig. B3: peaks b*, c*, f*, and g*).

BHT (17, 21(H), 22$R$, 32$R$, 33$R$, 34$S$ stereochemistry) was predominant in most samples, along with unsaturated and methylated versions of BHT. In the surface sediment from the coastal lagoons, Cubres East and West, we

identified two compounds comparable to BHT, however, in the MS$^2$ spectrum we observe the loss a methoxy moiety (32 Da, $CH_3OH$), as well as three hydroxyl moieties (Appendix B2, Fig. B4). We tentatively identify these compounds as methoxylated-BHTs. BHT-cyclitol ether (CE) was also abundant in both water column SPM and surface sediments (Fig. 2). In addition, we detected MeBHT-CE in surface sediments from Lake Verde, as well as an isomer where, based on the MS$^2$ spectrum, the methylation occurs on the cyclitol ether head group (Appendix

B3, Fig. B5). We putatively identify this isomer as BHT-MeCE. BHpentol, BHpentol-CE, BHhexol, and BHhexol-CE were also identified at our sites but only in the lake surface sediments.

Aminotriol, as well as isomers of aminotriol and unsaturated aminotriol, were abundant in the water column SPM and surface sediment of the lakes (Fig. 2). Methylated and acylated-versions of aminotriol were also identified,

but only in the surface





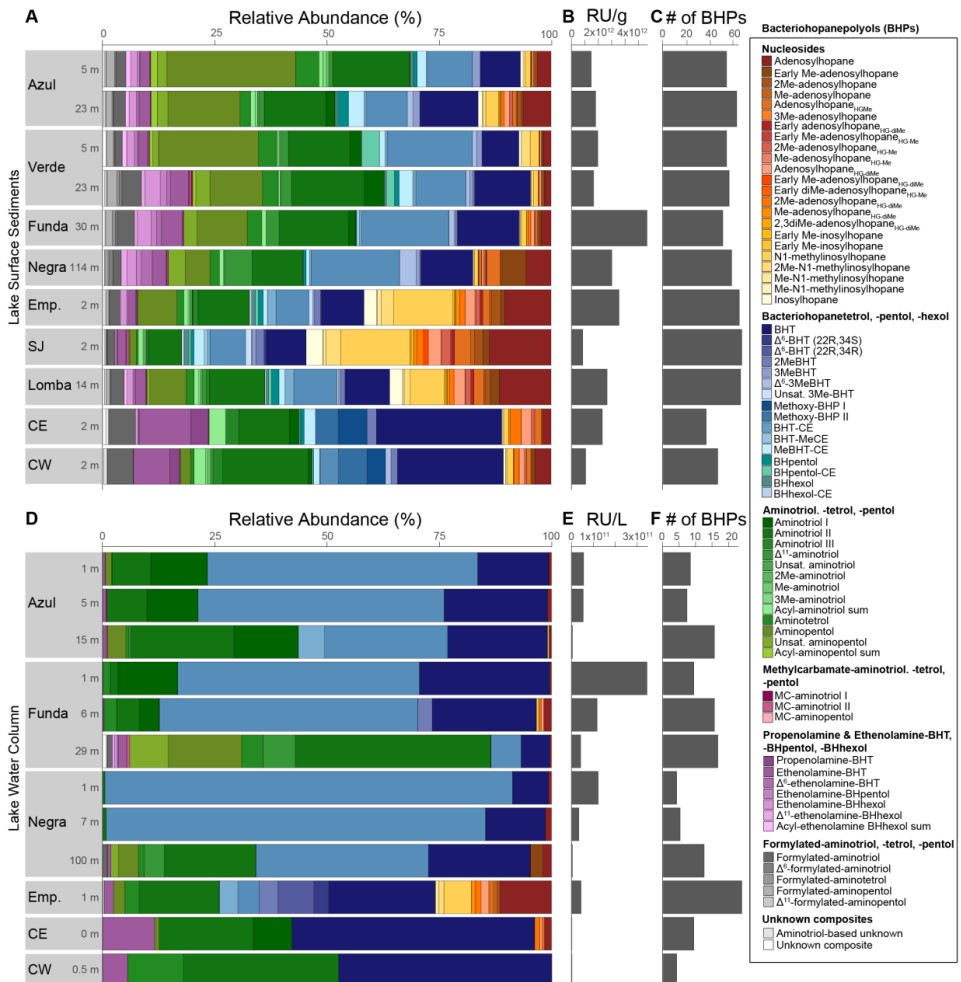

**Figure 2: (A) Relative abundance of all BHPs found in surface sediment samples labelled with the lake depth (meters) at which they were collected where Emp. = Empadadas, SJ = São Jorge, CE = Cubres East, and CW = Cubres West (all raw data is available in the supplementary material). (B) The response units (RU) of BHPs in the sediment samples normalized to g of dry sediment and (C) the number of BHPs identified in each sample. (D) Relative abundance of BHPs identified in water column samples from the different lakes and lagoons (E) with the total response units/L of water filtered and (F) number of BHPs shown.**

sediment of the lakes. In contrast to a previous study in a hyper-euxinic and meromictic lake, we do not observe any diunsaturated-aminotriols in the SPM or surface sediments at our study sites (O'Beirne et al., 2022).

Aminotetrol and aminopentol, compounds commonly associated with methanotrophic activity (Neunlist and Rohmer, 1985b; Zundel and Rohmer, 1985; Cvejic et al., 2000), were found in almost all of the water column SPM and surface sediments. However, acylated versions of aminopentol ($C_{14:0}$, $C_{15:0}$, and $C_{16:0}$) were only



observed in the surface sediments of Azul and Verde. Two isomers of methylcarbamate (MC)-aminotriol and MC-aminopentol, compounds also associated with methanotrophs, were identified in water column SPM and surface

sediment after Rush et al. (2016). The recently described propenolamine-BHT, ethenolamine-BHT, ethenolamine-BHpentol, and ethenolamine-BHhexol (Fig. 3) were identified in our samples after Hopmans et al. (2021). In addition, we putatively identified unsaturated versions of ethenolamine-BHT (Appendix B4, Fig. B6) and ethenolamine-BHhexol (Fig. 3) with the double bond likely occurring at the $\Delta^6$ and $\Delta^{11}$ positions, respectively, based on the retention times. Acylated versions of ethenolamine-BHhexol ($C_{15:0}$, $C_{16:0}$, and $C_{17:0}$) were also

observed in lake surface sediments.

In Hopmans et al. (2021) several unknown composite BHPs were described with an assigned elemental composition (AEC) of $C_{36}H_{64}O_4N^+$ (*m/z* 574.483) and $C_{36}H_{64}O_5N^+$ (*m/z* 590.483), respectively, but no structure was assigned. Here, we detected these novel BHPs in sediment from Lake Verde (Fig. 3, peaks j and m) based on

the diagnostic loss of 28 Da (CO) in the $MS^2$ spectra. We also identified the same compound with an additional hydroxy moiety on the side chain in the partial mass chromatogram of *m/z* 606.473 (elemental composition (EC) $C_{36}H_{64}O_6N^+$; Fig. 3, peak o). We propose that these novel composite BHPs are a series of formylated-aminoBHPs: formylated-aminotriol (peak j), formylated-aminotetrol (peak m) and formylated-aminopentol (peak o). The proposed structure for formylated-aminopentol is shown in Fig. 3f with the diagnostic fragmentations indicated.

In addition, we observe unsaturated versions of formylated-aminotriol (Fig. 3b & Appendix B5, Fig. B7), formylated-aminotetrol (based on the $MS^1$ and retention time), and formylated-aminopentol with the double bond likely occurring at the $\Delta^6$ position for formylated-aminotriol and $\Delta^{11}$ position for formylated-aminotetrol and formylated-aminopentol. In the mass chromatogram for *m/z* 572.467, we also observe a later-eluting peak ($C_{36}H_{62}O_4N^+$; Fig. 3b and d, peak l). Based on the $MS^2$ spectrum, we propose that this is the same compound

identified by Elling et al. (2022) in nitrite-oxidizing bacteria. The authors describe an acetylated compound (*m/z* 656.495; $C_{40}H_{66}O_6N^+$), with the unique loss of 44 Da ($CO_2$), the loss of two hydroxyl moieties, and an additional loss of 17 Da ($NH_3^+$). This compound is likely a cyclized form of formylated-aminotriol, and we putatively identify it as oxazinone-aminotriol (Appendix B6).

In addition to the discussed $\Delta^6$-ethenolamine-BHT, we observed a later eluting peak at 21.88 mins (Fig. 3a, peak c) in the mass chromatogram of *m/z* 586.483 ($C_{37}H_{64}O_4N^+$; $\Delta$ ppm 0.72). The $MS^2$ spectra is characterized by the distinct loss of 28 Da and a loss of 31 Da ($CNH_5$). Based on the elemental composition there appears to be an additional DBE (double bond equivalent) in the structure, but there is no indication that the unsaturation is located in the core structure as the lower mass range of the





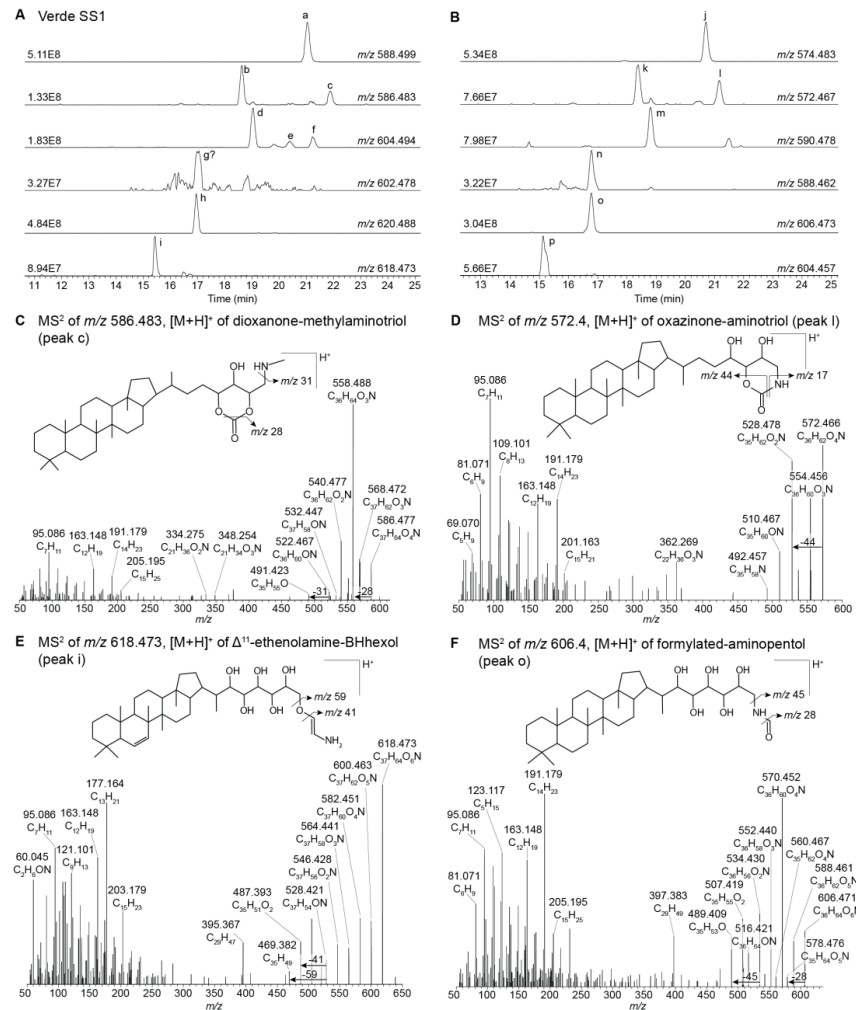


**Figure 3: (A) Partial mass chromatograms of ethenolamine-BHT (peak a), $\Delta^6$-ethenolamine-BHT (peak b), dioxanone-methylaminotriol (peak c), ethenolamine-BHpentol (peak d), isomers of methylcarbamate aminotriol (peaks e and f), a potential peak for unsaturated ethenolamine-BHpentol (peak g), ethenolamine-BHhexol (peak h), and $\Delta^{11}$-ethenolamine-BHhexol (peak i). (B) Partial mass chromatograms**
**of formylated-aminotriol (peak j), $\Delta^6$-formylated-aminotriol (peak k), oxazinone-aminotriol (peak l), formylated-aminotetrol (peak m), $\Delta^{11}$-formylated-aminotetrol (peak n), formylated-aminopentol (peak o), and $\Delta^{11}$-formylated-aminopentol (peak p). MS$^2$ spectrum of (C) the dioxanone-methylaminotriol in m/z 586.483 (peak c) and (D) the oxazinone-aminotriol composite (peak l) with tentative structures shown. MS$^2$ spectrum of (E) [M+H]$^+$ $\Delta^{11}$-ethenolamine-BHhexol and (F) [M+H]$^+$ formylated-aminopentol with tentative**
**structures shown. The chromatograms are from a surface sediment sample from Lake Verde.**

fragmentation spectrum is similar to what is expected for BHT. We therefore assume one of the functionalities is part of a cyclized structure and propose this BHP is a formylated aminotetrol, where the formic acid is part of a cyclic structure and the terminal amino group is methylated. The proposed structure with key fragmentations is shown in Fig. 3c. We tentatively identify this compound as a dioxanone-methylaminotriol (Appendix B7).



### 3.2 Distribution of BHPs depend on environmental conditions in lakes and coastal lagoons


Based on the NMDS visualization, the more complex BHP distributions in the surface sediments allow these sampling sites to be placed into three lake types (Fig. 4): deep lakes (i.e., Azul, Verde, Funda, and Negra), shallow lakes (i.e., Empadadas, São Jorge, Lomba), and coastal lagoons (i.e., Cubres East and Cubres West). We find a significant difference both between the sample sites (R = 0.89, $p$ < 0.01) and the sample site-types (R = 0.72, $p$ <

0.001). The variability between our sites, particularly in the surface sediments, is reflected in the overall differences in BHP distributions. In Azul and Verde surface sediments collected near the shore and in the deepest part of the lake show comparable BHP distributions. We also observe comparable BHP distributions in sample replicates for lakes Azul, Empadadas, and Negra, thus we will only discuss the average BHP values for the replicate sites. In the water column NMDS plot (Fig. 5) we observe the bottom water samples cluster closer

together, whereas the samples from the surface water and chemocline overlap. Using the ANOSIM test, we find there is no significant difference in BHP distributions between the surface water, chemocline, and bottom water of these lakes (R = 0.04, $p$ = 0.32) or between the water columns of different sites (R = 0.06, $p$ = 0.35).

The deep lakes are monomictic and undergo stratification from June-October, with hypoxic and even anoxic

conditions occurring during the summer months in the hypolimnion (Gonçalves, 2008; Raposeiro et al., 2018). These lakes contain a high abundance of BHT, BHT-CE, and aminotriol, particularly in the surface waters and sediment samples (Fig. 2 & Tables A3-A4). Previous studies that identified BHPs in lake sediments also described a high abundance of BHT, BHT-CE, and aminotriol (Farrimond et al., 2000; Talbot and Farrimond, 2007). Similarly, BHP distributions in the water column of a hypereutrophic, meromictic lake, Lake Mahoney (Canada),

are dominated by BHT, unsaturated BHT, aminotriol, and BHT-CE in the surface waters and BHT, diunsaturated aminotriol, and aminotriol in the surface sediments (O'Beirne et al., 2022). In the Azores, the deep lakes are further distinguished by the presence of amino-containing BHPs in the surface sediments (Fig. 2a) that cluster together in the NMDS plot (Fig. 4), including: aminotetrol, aminopentol, MC-aminopentol, ethenolamine-BHpentol, and ethenolamine-BHhexol. This is also reflected in the BHP distributions of the bottom water SPM

samples collected in June 2018 (Fig. 6), in which we observe amino-containing BHPs (i.e., aminopentol, ethenolamine-BHhexol, aminopentol, formylated-aminopentol) in samples collected below the oxycline. A higher abundance of penta- and hexa-functionalized BHPs, including aminopentol, were previously described in eutrophic lakes relative to lakes with low primary productivity (Farrimond et al., 2000; Talbot and Farrimond, 2007).


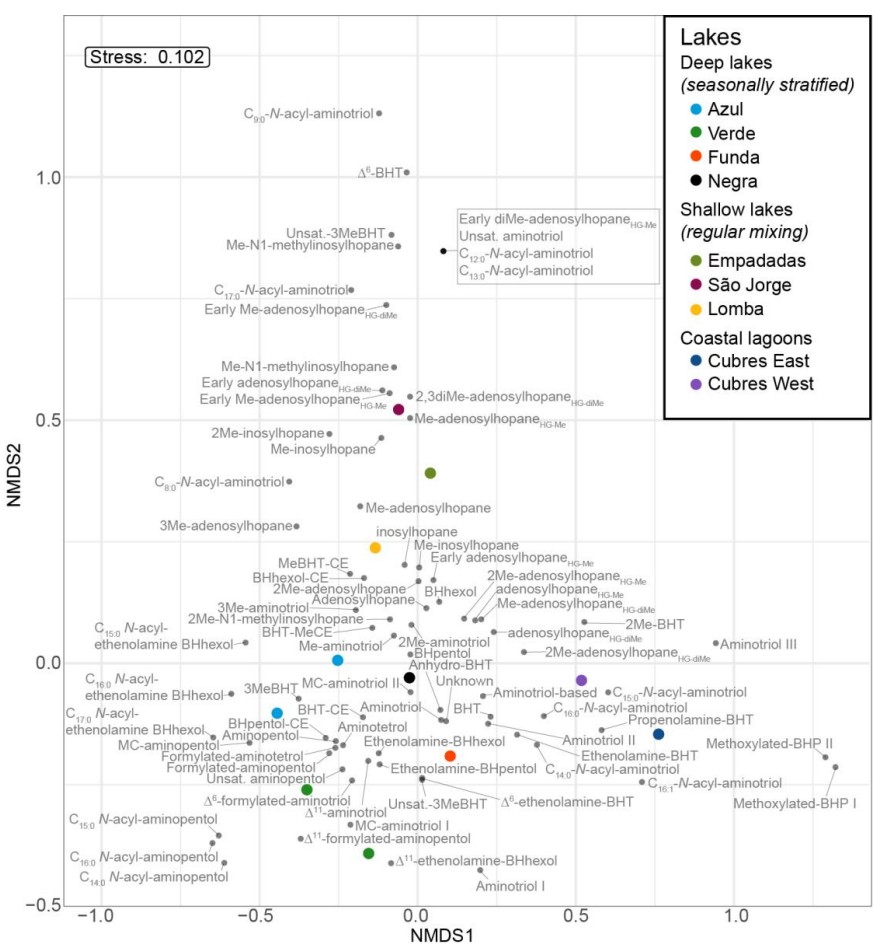

**Figure 4: NMDS visualization of BHP distributions in surface sediments from the different lakes and lagoons analyzed in this study.**

Although the shallower lakes might also undergo a period of stagnation, more regular mixing likely leads to oxygenation of the surface sediments. The shallow lakes are characterized by a higher abundance and diversity of nucleoside BHPs in both the surface sediment and water column SPM (Fig. 7). In addition, Me-inosylhopane, Me-N1-methylinosylhopane, and the early eluting adenosylhopane-type BHPs all cluster together near the shallow lakes, São Jorge and Empadadas, in the NMDS plot (Fig. 4). Nucleoside BHPs (i.e., adenosylhopane,

adenosylhopane-type, and 2Me-adenosylhopane-type) were previously reported in the sediment samples of two lakes, Loch Ness (Scotland) and Lake Nkunga (Kenya), that were associated with high levels of terrestrial input (Talbot and Farrimond, 2007). Loch Ness and Lake Nkunga also contained relatively high levels





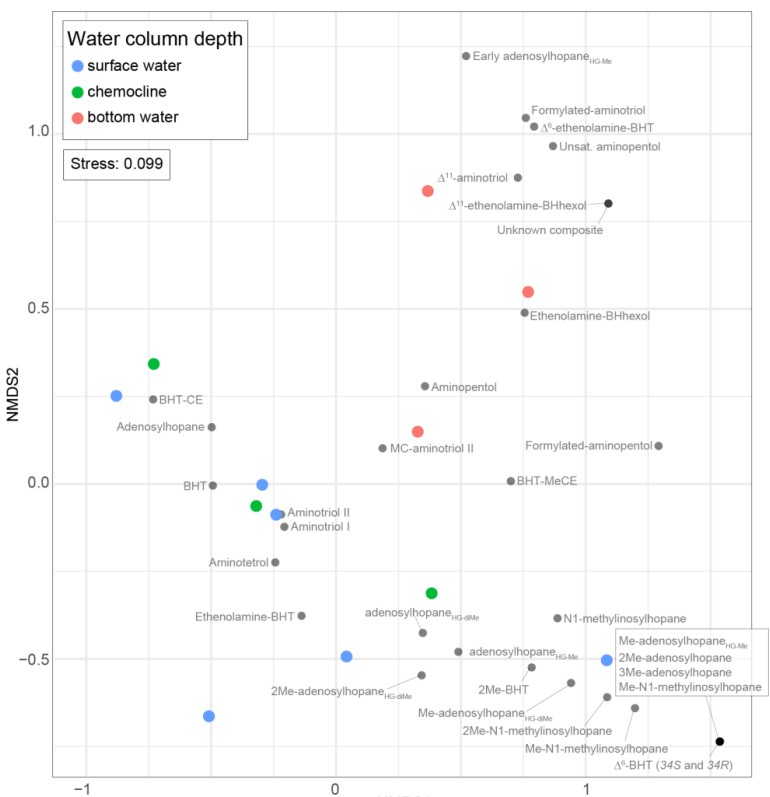

**Figure 5: NMDS visualization of BHP distributions between different water column samples from lakes Azul, Empadadas, Funda, and Negra, as well as Cubres East and West. Note: For Empadadas, Cubres East, and Cubres West we only have surface water column samples since the sampling locations were less than 3.5 m deep.**

of BHT, aminotriol, and BHT-CE, and even aminopentol (Talbot and Farrimond, 2007). Similar BHP distributions are observed in the sediments of Empadadas, São Jorge, and Lomba.


Finally, Cubres East and West are shallow, coastal lagoons that are influenced by both marine and freshwater inputs. Cubres East (salinity 9.7 ppt) and West (salinity 23.8 ppt) are defined by a high relative abundance of ethenolamine-BHT and propenolamine-BHT, and by the presence of two methoxylated BHPs. In contrast to our freshwater sites, BHT-CE is mainly absent from the coastal lagoons (except for in the surface sediments of Cubres

West). Similarly, in land-to-sea transects from the Yangtze (China) and Yenisei (Siberia) Rivers, BHT-CE was more abundant in rivers, estuaries, and the coast relative to the open water (Zhu et al., 2011; De Jonge et al., 2016). Marine samples and fjord samples, appear to predominantly consist



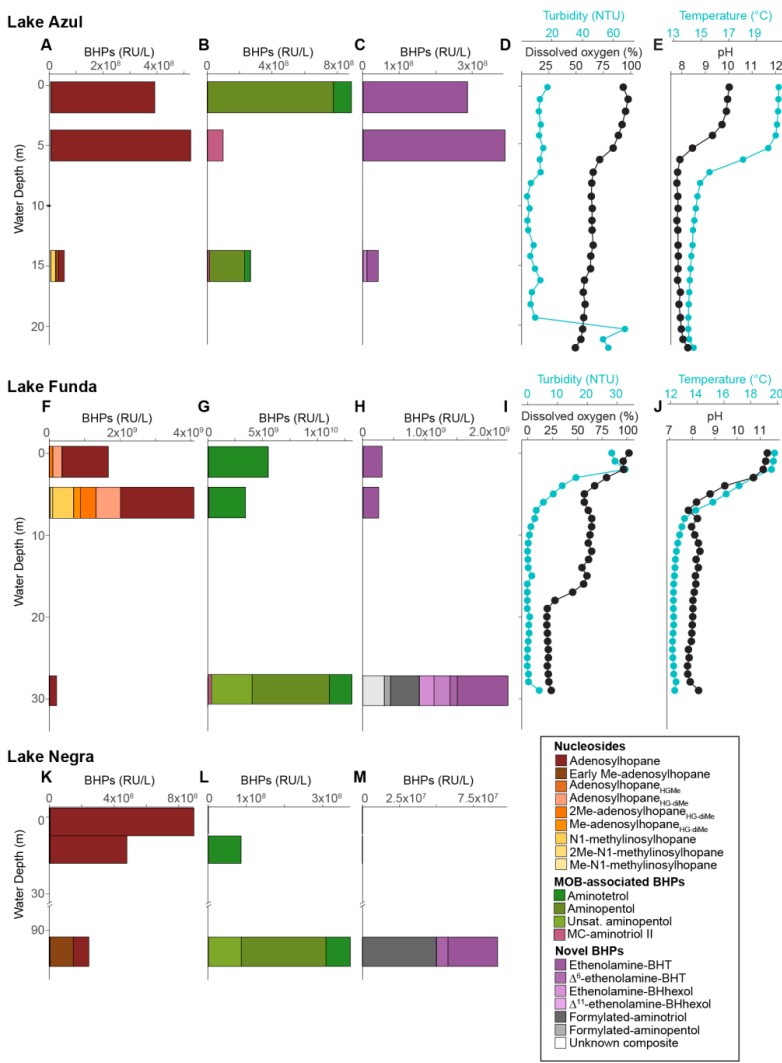

**Figure 6: Water column profiles for Lakes Azul (A-E), Funda (F-J), and Negra (K-M). Plots A, F, and K show the relative units/liter (RU/L) of nucleoside BHPs identified in the water column. Plots B, G, and L show the relative units/liter (RU/L) of BHPs associated with methane-oxidizing bacteria found in the water column. Plots C, H, and M display the relative units/liter (RU/L) of ethanolamine and recently identified composite BHPs identified in the water column. D and I show the turbidity (blue) and dissolved oxygen (pink) from the water column at the time of sampling in June 2018. Similarly, E and J display the temperature (blue) and pH (pink) profiles of the water column at the time of sampling for both Lakes Azul and Funda.**



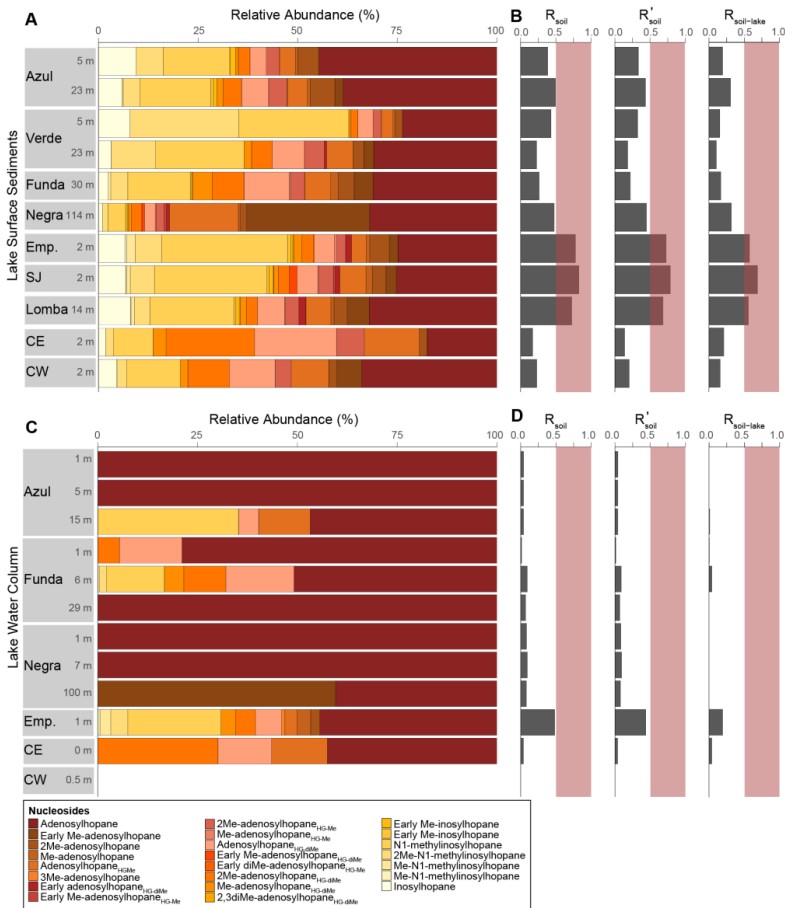


**Figure 7: (A)** Relative abundance of nucleosides identified in the surface sediment samples for all Azorean lakes (where Emp = Empadadas; SJ = São Jorge, CE = Cubres East; CW = Cubres West) with the **(B)** $R_{soil}$ (see equation 1), $R'_{soil}$, and $R_{soil\text{-}lake}$ calculated for each sample. **(C)** The distribution of nucleosides in water
column samples and their respective **(D)** $R_{soil}$, $R'_{soil}$, and $R_{soil\text{-}lake}$ values are also shown. The red bars indicate the terrestrial end-member values.



of BHT and aminotriol (Farrimond et al., 2000; Zhu et al., 2011; Kusch et al., 2021a). In the estuary and coastal samples from the Yangtze River drainage basin, the BHP distributions consist of BHT, 2MeBHT, aminotriol,

aminotetrol, aminopentol, nucleoside BHPs, and BHT-CE (Zhu et al., 2011). In general, we observe a similar distribution of BHPs in Cubres East and West.

Aminotriol and BHT were the most abundant BHPs at all of our sample sites. The presence of more diverse and abundant amino-BHPs in the deeper and seasonally stratified lakes allow us to distinguish them from the shallower

lakes that contain a higher abundance of nucleoside BHPs. Finally, the lower diversity of BHPs and unique presence of methoxylated BHPs in Cubres East and West, separates the marine influenced sites from the rest of the lakes in this study.

### 3.3 Diversity of BHPs in a methanotroph-methylotroph co-culture enriched from a lake water column

Previously, certain BHPs identified in MOB cultures were also detected in lake settings (Talbot et al., 2001; Talbot

and Farrimond, 2007; Rush et al., 2016; Neunlist and Rohmer, 1985b, c; Cvejic et al., 2000; O'Beirne et al., 2022). However, neither methanotrophs nor lacustrine settings have been analyzed since the recent methodological advancements have expanded BHP identification (Hopmans et al., 2021).

Here, we investigated the BHP composition of a lacustrine methanotroph-methylotroph (*Methylobacter-*

*Methylotenera*) co-culture isolated from Lacamas Lake, WA, U.S.A. (van Grinsven et al., 2020). BHPs (i.e., aminotriol, 3Me-aminotriol, MC-aminotriol, aminotetrol, 3Me-aminotetrol, MC-aminotetrol, aminopentol, unsaturated-aminopentol, 3Me-aminopentol, MC-aminopentol) have been previously detected in cultures of gammaproteobacterial Type I MOB *Methylobacter* (Osborne, 2015; Rush et al., 2016). Our aim was to potentially identify novel BHPs from a lacustrine enrichment dominated by *Methylobacter*. This enrichment is also abundant

in the methylotroph *Methylotenera* (van Grinsven et al., 2020). In order to rule out the synthesis of BHPs by this methylotroph, we investigated its genomic capacity to synthesize BHPs by searching for the key gene responsible for the formation of hopane, and therefore BHP (i.e., squalene hopene cyclase). A protein blast search (NCBI) was performed with the sequence of the squalene hopene cyclase gene of *Bradyrhizobium japonicum* as a query (accession number WP_038942977.1), which did not lead to any hits in the genomes of *Methylotenera* species.

In addition, we analyzed the BHP composition of *Methylotenera mobilis,* a closely related strain to the methylotroph also present in the co-culture, which was available in the DSMZ culture collection. Analysis of the *M. mobilis* biomass did not lead to the detection of BHPs.

On the other hand, in the *Methylobacter-Methylotenera* co-culture we identified 22 BHPs (Table A4), with the

most abundant being aminotriol (43 %) and aminopentol (43 %) (Fig. 8). Aminotetrol (6 %) was also observed, along with the unsaturated and acylated forms of aminotriol and aminopentol. Although present in low abundance, we identified MC-aminotriol, MC-aminopentol, ethenolamine-BHT, ethenolamine-BHhexol, formylated-aminotriol, and formylated-aminopentol (Fig. 8, Table A4). Low concentrations of nucleoside BHPs, i.e. adenosylhopane and 2Me-adenosylhopane$_{HG-diMe}$, were also present. Finally,



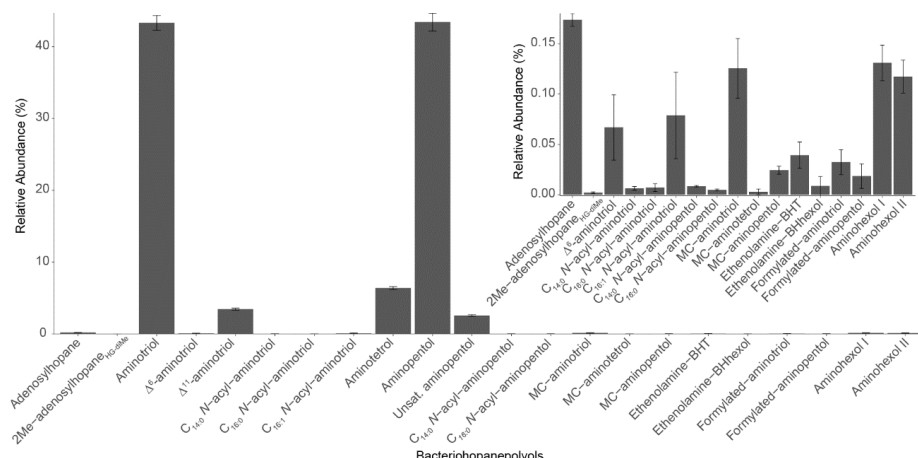


**Figure 8: BHP profile of *Methylobacter-Methylotenera* co-culture (n=3). Note: the BHPs present in low abundance are shown in the inset figure.**

we detected two novel BHPs ($m/z$ 594; $C_{35}H_{64}O_6N^+$; Fig. 9) where the $MS^2$ spectra show the consecutive loss of five hydroxyl moieties and an additional loss of 29 Da ($CH_3N$). Based on the $MS^2$ spectra and the elemental

composition, we tentatively identify these compounds as aminohexol BHPs (Appendix B8). Notably, 3β-methylated-BHPs are absent in the co-culture. Based on past culture studies, aminopentol and aminotetrol are associated with Type I and Type II MOB, respectively (Neunlist and Rohmer, 1985b, c; Cvejic et al., 2000; Talbot et al., 2001; van Winden et al., 2012). The high relative abundance of aminopentol relative to aminotetrol observed in the BHP distribution of the *Methylobacter-Methylotenera* co-culture analyzed in this study is similar to that of

other Type I MOB (Neunlist and Rohmer, 1985c; Talbot et al., 2001; van Winden et al., 2012; Rush et al., 2016; Kusch and Rush, 2022). The novel BHPs detected in the co-culture, however, are reported for the first time in this study. We conclude that the suite of BHPs identified in the co-culture, including the novel ethenolamine-BHPs, formylated-aminoBHPs, and aminohexols, are attributed to the MOB *Methylobacter* present, and thus can be potentially considered as markers of aerobic methanotrophs in lake settings.

**3.4 BHPs as biomarkers for aerobic methanotrophs in lakes**

Aminotetrol was present in all of the Azorean samples, except for in the surface water of Negra (1 m depth), at 5 m depth in Azul, and in the water column and sediment of Cubres East (Fig. 2). Similarly, aminopentol is observed in the water column and sediment of Azul and Verde (except for at a depth of 5 m in the water column of Azul). In Funda and Negra, aminopentol only occurs in the deepest part of the water column and in the sediment samples.

For Empadadas and Cubres East, aminopentol occurs both in the water column and the sediment, but it is only observed in the sediment sample from Cubres West. Similarly, it is found in both sediment samples from Lomba and São Jorge. Both aminopentol and aminotetrol are also produced in small





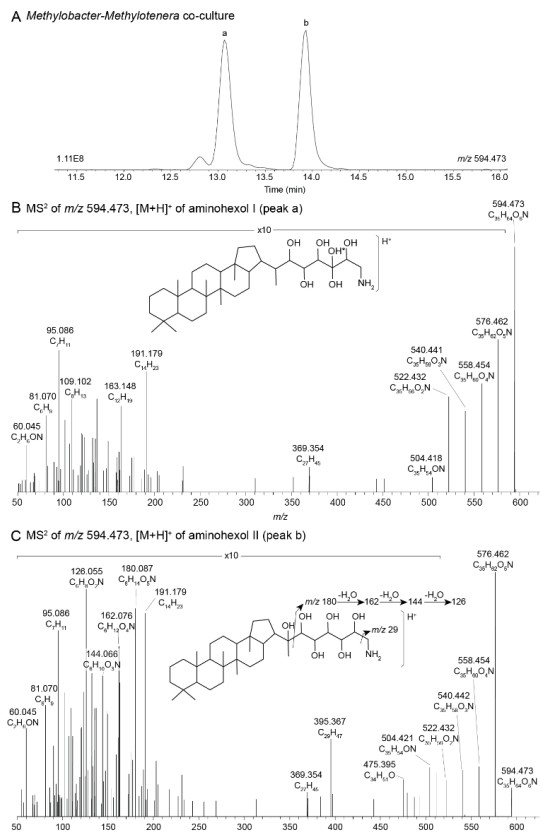

**Figure 9: (A) Partial mass chromatograms of aminohexol I (peak a) and II (peak b) BHPs from the *Methylobacter-Methylotenera* co-culture with. MS$^2$ spectra for (B) aminohexol I and (C) aminohexol II and the proposed structures. The asterisk (*) indicates arbitrary placement of the hydroxyl group on the side chain.**

amounts by sulfur-reducing bacteria (SRB; Blumenberg et al., 2006). As SRB also synthesize relatively higher

amounts of aminotriol, ratios of aminopentol and aminotetrol to aminotriol have been used to determine the origin

of the penta- and hexa-functionalised amino-BHPs (Blumenberg et al., 2006, 2009, 2012). The absence of anoxic

conditions in the lake water column would rule out a major contribution of SRB-produced BHPs during the time

of sampling. This is confirmed by the high aminopentol:aminotriol (water column: 0 (aminopentol absent) to 0.37

and sediment: 0.01 to 1.67) and aminotetrol:aminotriol (water column: 0 (aminotetrol absent) to 1.30 and

sediment: 0 (aminotetrol absent) to 0.38) ratios in both water column and sediment samples. Recently two species

of nitrite-oxidizing bacteria were shown to produce aminopentol in minor amounts under certain growth

conditions (Elling et al., 2022), potentially contributing to the aminopentol observed in the surface waters of Lake

Azul, Empadadas, and Cubres East. However, in the oxygen-depleted bottom waters of Azul, Funda, and Negra,

MOB are likely the primary producers of aminotetrol and aminopentol. A relatively higher abundance of

aminopentol than aminotetrol in the water column and sediment samples would suggest a higher abundance of

Type I than Type II MOB, in accordance with previous studies in both dimictic and meromictic lakes (Hanson

and Hanson, 1996; Oswald et al., 2015, 2016; Guggenheim et al., 2020).



We observe unsaturated aminopentol (van Winden et al., 2012; Wagner et al., 2014) in the bottom water samples

of Lake Funda and Negra (Fig. 6), which could be an adaptation by MOB to colder water temperatures (Bale et al., 2019). We also observe unsaturated aminotriol in the bottom water of Azul (15 m, 14 °C), Funda (29 m, 12 °C), and Negra (100 m; temperature data unavailable). $C_{16:0}$ N-acyl-aminopentol was reported in trace amounts near a terrestrial methane seep (Hopmans et al., 2021). $C_{16:0}$ N-acyl-aminopentol, as well as $C_{14:0}$ and $C_{15:0}$ N-acyl-aminopentol are present in sediment samples from Azul and Verde, whereas in Lomba and Empadadas we only

observe $C_{16:0}$ N-acyl-aminopentol in the sediment (Fig. 2). In the NMDS plot of the surface sediments, the acylated-aminopentols group closely with BHPs related to MOB, aminotetrol and aminopentol (Fig. 4). In addition, $C_{14:0}$ and $C_{16:0}$ N-acyl-aminopentol are both detected in the *Methylobacter-Methylotenera* co-culture, confirming that MOB could be a potential source for these acyl-amino-BHPs in Lakes Azul, Funda, Lomba, and Empadadas.


MC-aminotriol and MC-aminopentol were produced by the *Methylobacter-Methylotenera* co-culture (Fig. 8), albeit in low abundance. However, this is the first report of ethenolamine-BHT, ethenolamine-BHhexol, formylated-aminotriol, and formylated-aminopentol in culture. So far, ethenolamine-BHPs and formylated-aminoBHPs were only identified near a terrestrial methane seep (Hopmans et al., 2021). Further environmental

and culture studies are needed to identify whether these are robust biomarkers for methanotrophs. In all of our samples, except for the water column of Negra at 1 and 7 m depths (Fig. 2), we detected ethenolamines, propenolamine, and/or formylated-aminoBHPs (Hopmans et al., 2021). Their widespread distribution would suggest multiple producers. Although we cannot identify a direct source for these compounds, an NMDS analysis shows ethenolamine-BHpentol, ethenolamine-BHhexol, acylated-ethenolamine BHhexol ($C_{15:0}$, $C_{16:0}$, $C_{17:0}$),

formylated-aminotetrol, and formylated-aminopentol clustering with MC-aminopentol, aminopentol, aminotetrol and the acylated-aminopentols (Fig. 4). These aminoBHPs are primarily associated with surface sediments from Azul, Verde, Funda, and Negra and the bottom water from Funda and Negra (Fig. 2 & 6). Azul, Verde, Funda, and Negra all stratify during the summer months and the bottom water in these lakes range from suboxic to anoxic conditions (Gonçalves et al., 2018). This would suggest that ethenolamines, propenolamine, and/or formylated-

amino BHPs are being produced under suboxic to anoxic conditions and could be associated with MOB.

BHPs with a 3β-methylation, particularly 3MeBHT and 3Me-aminotriol are often linked to methane oxidation (Neunlist and Rohmer, 1985b; Zundel and Rohmer, 1985; Cvejic et al., 2000). However, 3β-methyl-BHPs are not exclusively produced by MOB (Zundel and Rohmer, 1985; Welander and Summons, 2012). BHPs with a 3β-

methylation are also produced by other bacteria in the phyla Alpha-, Beta-, and Gamma-proteobacteria, Cyanobacteria, Acidobacteriota, and Nitrospirae (Zundel and Rohmer, 1985; Sinninghe Damsté et al., 2017; Elling et al., 2022). In addition, 3β-methylation is crucial for bacterial cell survival in the late stationary phase and thus could have a broad taxonomic distribution (Welander and Summons, 2012). We observe 3MeBHT in all sediment samples (except for Cubres East and West; Fig. 2), however, it was not detected in our water column

samples. In contrast, 3Me-aminotriol only occurs in sediment samples from Lomba, Cubres West, São Jorge, Empadadas, Verde, and Azul. The absence of 3Me-BHPs in SPM samples suggests that at the time of sampling the bacteria responsible for 3β-methylation were not abundant in the water column. Future work will need to





evaluate annual changes in both BHP distributions and bacterial communities in the water column to confirm the primary producers of 3Me-BHPs in lakes.


AminoBHPs with a methylcarbamate terminal group (MC-aminoBHPs) were proposed as biomarkers for Type I MOB in marine settings based on culture samples (Rush et al., 2016), but so far, the occurrence of MC-aminoBHPs in lakes has not been evaluated. MC-aminotriol has been observed in nitrite-oxidizing bacteria (Elling et al., 2022). The highest rates of methane-oxidation in seasonally stratified lakes typically occurs at or directly

below the chemocline (Rudd et al., 1976; Hanson, 1980; Harrits and Hanson, 1980; Oswald et al., 2015). In the Azorean lakes, MC-aminotriol was found in the oxycline and bottom water of both Lake Azul and Funda (Fig. 6). In Azul and Funda we observe MC-aminotriol near the oxycline, which could also be attributed to nitrite-oxidizing bacteria (Elling et al., 2022). We also observe MC-aminotriol in sediment samples from Verde, Negra, São Jorge, Lomba, Cubres East, and Empadadas. The co-occurrence of MC-aminopentol with 3MeBHT and/or 3Me-

aminotriol in sediment samples from Verde, Negra, Lomba, and Empadadas could indicate a common MOB source. Both Lomba and Empadadas are relatively shallow lakes with a fully oxygenated water column, which means the primary site of methane oxidation occurs in the surface sediment and not in the water column (Hanson and Hanson, 1996). The different MC-aminoBHP distributions in the sediment samples could indicate that we were either not sampling during the peak period of methane oxidation in the lake, there was a shift in the MOB

community, or methane oxidation was primarily occurring at the sediment surface.

Based on the distributions of aminoBHPs and MC-aminoBHPs, MOB are likely present, albeit in low-abundance, at all Azorean sites. In the deeper lakes (i.e., Azul, Funda, and Negra), methane oxidation is likely occurring either at or below the chemocline which would account for the diverse aminoBHPs observed in the water column (Fig.

6). Low $\delta^{13}C$ values in bulk sediment samples from Lake Funda and primarily reducing conditions associated with the bottom water of the lake, would suggest that methanogenesis is occurring in the bottom water and/or surface sediment of Lake Funda during the summer months (Richter et al., 2022). Similarly, proxy records suggest that eutrophication in Lake Azul contributed to hypoxic conditions in the bottom water, resulting in a shift in the chironomid community (Raposeiro et al., 2017). In both lakes Funda and Azul, an increase in organic matter

deposition combined with low oxygen conditions in the bottom water during the summer months would support methane production, and therefore, a community of methane-oxidizing bacteria in the water column of these lakes. A similar distribution of BHPs associated with methane-oxidizing bacteria (i.e. aminopentol, unsaturated aminopentol, and aminotetrol) are found in high abundance in the methane-rich environments of floodplains in the Amazon basin and wetlands in the Congo River basin (Wagner et al., 2014; Spencer-Jones et al., 2015). In

contrast, in the shallow lakes and the coastal lagoons methane oxidation might occur both in the water column and in the oxygenated sediment. Further molecular and culture studies, however, are needed to validate these conclusions.

### 3.4 Autochthonous vs. allochthonous source of nucleoside BHPs

A recent study suggests that some nucleoside BHPs may be *in situ* produced in the chemocline or under

anoxic/euxinic conditions in marine water columns (Kusch et al., 2021b). To the best of our knowledge, no studies have investigated nucleoside BHP production in lacustrine environments. Nucleoside BHPs were detected in all





water column and sediment samples except the water column of lagoon Cubres West (salinity 23.8 ppt; Fig. 7). The distribution of nucleoside BHPs in the sediment of Lake Empadadas is reflected in the BHPs distribution of its surface water, suggesting that in shallow, well-mixed lakes nucleoside BHPs like adenosylhopane and N1-

methylinosylhopane are either derived from production in the water column or surface water run-off (Cooke et al., 2008; Xu et al., 2009; Rethemeyer et al., 2010). In contrast, in Lake Funda we observe an increase in adenosylhopane, adensylhopane$_{HG-diMe}$, 2Me-adenosylhopane$_{HG-diMe}$, Me-adenosylhopane$_{HG-diMe}$, N1-methylinosylhopane, 2Me-N1-methylinosylhopane, and Me-N1-methylinosylhopane at the chemocline (6 m depth). Of these nucleoside BHPs, only adenosylhopane, adenosylhopane$_{HG-diMe}$, and 2Me-adenosylhopane$_{HG-diMe}$

are present in the surface water (1 m) of Lake Funda. A pycnocline associated with the chemocline in the Azores lakes could result in the accumulation of adenosylhopane, adenosylhopane$_{HG-diMe}$, and 2Me-adenosylhopane$_{HG-diMe}$, however, the lack of Me-adenosylhopane$_{HG-diMe}$, N1-methylinosylhopane, 2Me-N1-methylinosylhopane, and Me-N1-methylinosylhopane in the surface water of Lake Funda, could also indicate *in situ* production of these compounds in the water column. Kusch et al. (2021b) found that adenosylhopane, N1-methylinosylhopane, and

2Me-N1-methylinosylhopane were produced within the chemocline and under anoxic/euxinic conditions in marine settings. In addition, adenosylhopane is separate from the rest of the nucleosides in the NMDS plot of only water column samples (Fig. 5), falling in between samples collected near the chemocline and the surface water. N1-methylinosylhopane is also observed in the bottom water of Lake Azul, in addition to adenosylhopane$_{HG-Me}$ and adenosylhopane$_{HG-diMe}$, suggesting that these nucleosides are likely produced *in situ*. The primary producers

of nucleoside BHPs, however, remain unconstrained and are likely associated with a wide range of bacteria. Adenosylhopane, in particular, is produced as an intermediate product in BHP synthesis (Bradley et al., 2010) and has the potential to be sourced from diverse bacteria, including purple non-sulfur bacteria (Neunlist and Rohmer, 1985a; Neunlist et al., 1988; Talbot et al., 2007), nitrifying bacteria (Seemann et al., 1999; Talbot et al., 2007; Elling et al., 2022), and nitrogen-fixing bacteria (Bravo et al., 2001; Talbot et al., 2007).

**3.5 Implications for the R$_{soil}$ proxy**

The ratio of nucleoside BHPs to BHT (R$_{soil}$ index, Eq. 1; Zhu et al., 2011) and the revised soil index for high latitudes (R′$_{soil}$, Eq. 2; Doğrul Selver et al., 2012) have previously been used as an indication of soil input to riverine and marine environments (Cooke et al., 2009; Sáenz et al., 2011; Taylor and Harvey, 2011; Zhu et al., 2011; Doğrul Selver et al., 2012, 2015; De Jonge et al., 2016; Kusch et al., 2019). However, the recent indications

that nucleoside BHPs may be *in situ* produced in marine (Kusch et al., 2021b) and lacustrine water columns (this study) could affect the application of the R$_{soil}$ proxy. R$_{soil}$ values of marine and lacustrine sediments from the tropics to Arctic range from 0.4-0.9 (Cooke et al., 2008; Pearson et al., 2009; Rethemeyer et al., 2010; Kusch et al., 2021a), and values between 0.5-0.8 have been employed as a tentative terrestrial end-member (Zhu et al., 2011). Here we revise the R$_{soil}$ index to exclude nucleoside BHPs that might be produced in the lake water column

(i.e., adenosylhopane, early eluting adenosylhopane$_{HG-Me}$, Me-adenosylhopane$_{HG-diMe}$, N1-methylinoslyhopane, 2Me-N1-methylinsolyhopane, Me-N1-methylinoslyhopane). We call this new index, R$_{soil-lake}$. This ratio is significantly correlated with R$_{soil}$ (R = 0.94, $p<0.0001$) and R′$_{soil}$ (R = 0.60, $p<0.0001$). Further, this ratio is positively correlated with TOC/TN (R = 0.81, $p<0.005$; Fig. C1), where TOC/TN values between 4 to 10 reflect organic matter derived from lake algae and values higher than 20 indicate an increased source of vascular land

plants (Meyers, 2003). All of the sediment TOC/TN values fall between 7 and 11, indicating predominantly *in*





*situ* derived organic matter which is in accordance with the high levels of primary productivity observed in all of the lakes (Cordeiro et al., 2020).

At all of the sites, we observe lower $R_{soil-lake}$ values in the water column than the sediment. As our water column
SPM samples were collected in the dry season in the Azores (April-August; Hernández et al., 2016), we would anticipate low terrestrial inputs. In contrast, the BHPs detected in the sediments represent an integrated annual signal, and likely includes a larger input of terrestrial material during the wet season (September-March; Hernández et al., 2016). The $R_{soil-lake}$ values found in the sediments of the deeper lakes (i.e., Funda, Negra, and Azul) range from 0.17 to 0.32, and similarly have TOC/TN values between 8.1 and 8.3. The sediments of the
shallow lakes all contain relatively higher $R_{soil-lake}$ values (i.e., Empadadas = 0.57, São Jorge = 0.69, and Lomba = 0.56) and higher TOC/TN values (ranging from 10 to 11.4), which could indicate slightly higher inputs of terrestrial organic matter, particularly during the wet season. In the coastal lagoons, Cubres East and West, we observe low $R_{soil-lake}$ values both in the water column (0.04 Cubres East and 0 Cubres West) and sediment samples (0.21 Cubres East and 0.16 Cubres West). This could reflect fewer terrestrial inputs relative to the other lakes in
this study. Low TOC/TN values (7.08 Cubres East and 7.22 Cubres West) would also indicate a primarily *in situ* source of organic matter (Meyers, 2003), however, further work is needed to verify this.

Our results highlight that caution should be used to ensure that the nucleoside BHPs included in the $R_{soil}$ proxy are not produced *in situ*. In our current dataset, the $R_{soil-lake}$ index, which excludes nucleoside BHPs that likely
originate from the water column, appears to work for all of our Azorean sites, however additional work is needed to distinguish the primary source of nucleoside BHPs, and to expand the application of $R_{soil-lake}$. More detailed water column and catchment studies are needed to confirm whether these nucleoside BHPs are actually being produced within the lakes. Furthermore, similar comparison studies should be performed in marine settings.

**4 Conclusions**

This study highlights the diversity and complexity of BHP distributions and their interpretations in the lacustrine and coastal environments. We identified several novel BHPs that are being described for the first time in lacustrine and coastal settings, including unsaturated ethenolamine-BHPs and formylated-aminoBHPs. Further, we identified several nucleoside BHPs that could be produced in the water column of lacustrine settings, however, further work is needed to verify the source of these BHPs. Within lakes, aminotetrol, aminopentol, and MC-
aminopentol show potential as proxies for MOB. In addition, the recently identified ethenolamine-BHpentol, ethenolamine-BHhexol, formylated-aminotriol, and formylated-aminopentol are produced by a lacustrine methanotroph, *Methylobacter* sp., and within the Azorean lakes are associated with low-oxygen conditions. Additional studies focused on describing BHP distributions particularly in lacustrine settings and in culture studies, however, are needed to confirm whether certain BHPs can be used as biomarkers for biogeochemical
cycles or environmental conditions.





## Appendix A Supplementary Tables

**Table A1: BHPs identified in this study with the retention times (tr), assigned elemental composition (AEC), calculated exact mass ($M_{calc}$), and the $\Delta$ ppm ($\Delta$ ppm = (($M_{calc}$ − $M_{measured}$)/$M_{calc}$)\*$10^6$).**


| BHP | Sample | tr (min) | AEC | MS² ion | $M_{calc}$ (m/z) | $\Delta$ ppm | Fig. Reference | References |
|---|---|---|---|---|---|---|---|---|
| adenosylhopane | São Jorge SS2 | 21.71 | $C_{40}H_{64}O_3N_5$ | H⁺ | 662.500 | -0.11 | B2, peak a | Hopmans et al. 2021 |
| adenosylhopane$_{HG-Me}$ | São Jorge SS2 | 17.42 | $C_{41}H_{66}O_3N_5$ | H⁺ | 676.516 | 0.14 | B2, peak b* | this study |
| 2Me-adenosylhopane | São Jorge SS2 | 21.82 | $C_{41}H_{66}O_3N_5$ | H⁺ | 676.516 | -0.08 | B2, peak c | Hopmans et al. 2021 |
| Me-adenosylhopane, unknown isomer | São Jorge SS2 | 22.30 | $C_{41}H_{66}O_3N_5$ | H⁺ | 676.516 | 0.04 | B2, peak d | Hopmans et al. 2021 |
| adenosylhopane$_{HG-Me}$ | São Jorge SS2 | 22.90 | $C_{41}H_{66}O_3N_5$ | H⁺ | 676.516 | -0.18 | B2, peak e | Hopmans et al. 2021 |
| 3Me-adenosylhopane | Lomba SS2 | 23.76 | $C_{41}H_{66}O_3N_5$ | H⁺ | 676.516 | -1.16 | | Hopmans et al. 2021 |
| adenosylhopane$_{HG-diMe}$ | São Jorge SS2 | 17.60 | $C_{42}H_{68}O_3N_5$ | H⁺ | 690.532 | -0.81 | B2, peak f* | this study |
| Me-adenosylhopane$_{HG-Me}$ | São Jorge SS2 | 17.90 | $C_{42}H_{68}O_3N_5$ | H⁺ | 690.532 | -0.77 | B2, peak g* | this study |
| 2Me-adenosylhopane$_{HG-Me}$ | São Jorge SS2 | 22.96 | $C_{42}H_{68}O_3N_5$ | H⁺ | 690.532 | -0.04 | B2, peak h | Hopmans et al. 2021 |
| Me-adenosylhopane$_{HG-Me}$ | Empadadas SS7 | 23.62 | $C_{42}H_{68}O_3N_5$ | H⁺ | 690.532 | -0.81 | B2, peak i | Hopmans et al. 2021 |
| adenosylhopane$_{HG-diMe}$ | São Jorge SS2 | 25.15 | $C_{42}H_{68}O_3N_5$ | H⁺ | 690.532 | -0.25 | B2, peak j | Hopmans et al. 2021 |
| Me-adenosylhopane$_{HG-diMe}$ | São Jorge SS2 | 18.06 | $C_{43}H_{70}O_3N_5$ | H⁺ | 704.547 | 0.82 | B2, peak k* | this study |
| diMe-adenosylhopane$_{HG-Me}$ | Empadadas SS10 | 23.68 | $C_{43}H_{70}O_3N_5$ | H⁺ | 704.547 | -1.38 | B2, peak l* | this study |
| 2Me-adenosylhopane$_{HG-diMe}$ | São Jorge SS2 | 25.17 | $C_{43}H_{70}O_3N_5$ | H⁺ | 704.547 | -0.50 | B2, peak m | Hopmans et al. 2021 |
| Me-adenosylhopane$_{HG-diMe}$ | São Jorge SS2 | 26.09 | $C_{43}H_{70}O_3N_5$ | H⁺ | 704.547 | -0.40 | B2, peak n | Hopmans et al. 2021 |
| 2,3diMe-adenosylhopane$_{HG-diMe}$ | São Jorge SS2 | 26.14 | $C_{44}H_{72}O_3N_5$ | H⁺ | 718.563 | -0.10 | B2, peak o | Hopmans et al. 2021 |
| inosylhopane | Empadadas SS8 | 20.33 | $C_{40}H_{63}O_4N_4$ | H⁺ | 663.484 | -0.68 | B3, peak a | Hopmans et al. 2021 |
| 2Me-inosylhopane | Empadadas SS8 | 20.45 | $C_{41}H_{65}O_4N_4$ | H⁺ | 677.500 | -0.28 | B3, peak b* | this study |
| Me-inosylhopane, unknown isomer | Empadadas SS8 | 20.90 | $C_{41}H_{65}O_4N_4$ | H⁺ | 677.500 | -1.10 | B3, peak c* | this study |
| N1-methylinosylhopane | Empadadas SS8 | 23.91 | $C_{41}H_{65}O_4N_4$ | H⁺ | 677.500 | -0.96 | B3, peak d | Hopmans et al. 2021 |
| 2Me-N1-methylinosylhopane | Empadadas SS8 | 24.03 | $C_{42}H_{67}O_4N_4$ | H⁺ | 691.516 | -0.36 | B3, peak e | Hopmans et al. 2021 |
| Me-N1-methylinosylhopane, unknown isomer | Empadadas SS8 | 24.54 | $C_{42}H_{67}O_4N_4$ | H⁺ | 691.516 | -0.16 | B3, peak f* | this study |
| Me-N1-methylinosylhopane, unknown isomer | Empadadas SS8 | 24.99 | $C_{42}H_{67}O_4N_4$ | H⁺ | 691.516 | 1.39 | B3, peak g* | this study |
| BHT | São Jorge SS2 | 20.48 | $C_{35}H_{66}O_4N$ | NH₄⁺ | 564.499 | -0.17 | | |
| $\Delta^6$-BHT (34S) | São Jorge SS2 | 18.28 | $C_{35}H_{64}O_4N$ | NH₄⁺ | 562.483 | 0.34 | | |
| $\Delta^6$-BHT (34R) | Empadadas 001 | 18.43 | $C_{35}H_{64}O_4N$ | NH₄⁺ | 562.483 | 0.49 | | |
| 2MeBHT | São Jorge SS2 | 20.62 | $C_{36}H_{68}O_4N$ | NH₄⁺ | 578.514 | -0.14 | B4, peak a | |
| 3MeBHT | São Jorge SS2 | 22.26 | $C_{36}H_{68}O_4N$ | NH₄⁺ | 578.514 | 0.29 | | |
| $\Delta^6$-3MeBHT | Negra SS1 | 20.15 | $C_{36}H_{66}O_4N$ | NH₄⁺ | 576.499 | -0.97 | | |
| unsat. 3MeBHT | Sao Jorge SS1 | 22.07 | $C_{36}H_{66}O_4N$ | NH₄⁺ | 576.499 | -0.97 | | |
| Methoxylated-BHT I | Cubres SS3 | 21.71 | $C_{36}H_{68}O_4N$ | NH₄⁺ | 578.514 | -0.04 | B4, peak b | this study |
| Methoxylated-BHT II | Cubres SS6 | 22.3 | $C_{36}H_{68}O_4N$ | NH₄⁺ | 578.514 | -0.09 | B4, peak c | this study |
| BHT-CE | Verde SS1 | 17.15 | $C_{41}H_{74}O_8N$ | NH₄⁺ | 708.541 | -0.15 | B5, peak a | |



| | | | | | | | | |
|---|---|---|---|---|---|---|---|---|
| BHT-Me-CE | Verde SS1 | 17.25 | $C_{42}H_{76}O_8N$ | $NH_4^+$ | 722.557 | -1.64 | B5, peak b | this study |
| 3MeBHT-CE | Verde SS1 | 18.63 | $C_{42}H_{76}O_8N$ | $NH_4^+$ | 722.557 | -1.92 | B5, peak c | |
| Anhydro-BHT | Verde SS1 | 24.09 | $C_{35}H_{64}O_3N$ | $NH_4^+$ | 546.488 | -0.90 | | |
| | | | | | | | | |
| BHpentol | Verde SS1 | 18.59 | $C_{35}H_{66}O_5N$ | $NH_4^+$ | 580.494 | -1.77 | | |
| Bhpentol-CE | Verde SS1 | 15.91 | $C_{41}H_{74}O_9N$ | $NH_4^+$ | 724.536 | -0.08 | | |
| | | | | | | | | |
| BHhexol | Empadadas SS8 | 16.59 | $C_{35}H_{63}O_6$ | $H^+$ | 579.462 | -0.64 | | |
| BHhexol-CE | Empadadas SS8 | 14.39 | $C_{41}H_{74}O_{10}N$ | $NH_4^+$ | 740.531 | -0.63 | | |
| | | | | | | | | |
| aminotriol I | Cubres SS3 | 16.84 | $C_{35}H_{64}O_3N$ | $H^+$ | 546.488 | -0.14 | | |
| aminotriol II | Cubres SS3 | 17.52 | $C_{35}H_{64}O_3N$ | $H^+$ | 546.488 | -0.03 | | |
| aminotriol III | Cubres SS3 | 18.15 | $C_{35}H_{64}O_3N$ | $H^+$ | 546.488 | 0.83 | | |
| $\Delta^6$-aminotriol | *Methylobacter-Methylotenera* | 15.47 | $C_{35}H_{62}O_3N$ | $H^+$ | 544.472 | -0.64 | | |
| $\Delta^{11}$-aminotriol | Funda SS1 | 15.83 | $C_{35}H_{62}O_3N$ | $H^+$ | 544.472 | -0.22 | | |
| unsaturated aminotriol | Empadadas SS7 | 16.72 | $C_{35}H_{62}O_3N$ | $H^+$ | 544.472 | -1.47 | | |
| 2Me-aminotriol | Empadadas SS8 | 17.74 | $C_{36}H_{66}O_3N$ | $H^+$ | 560.504 | -1.11 | | |
| Me-aminotriol | Cubres SS6 | 18.01 | $C_{36}H_{66}O_3N$ | $H^+$ | 560.504 | 0.02 | | |
| 3Me-aminotriol | Cubres SS6 | 18.97 | $C_{36}H_{66}O_3N$ | $H^+$ | 560.504 | 0.32 | | |
| $C_{8:0}$-*N*-acyl-aminotriol | São Jorge SS1 | 25.01 | $C_{43}H_{78}O_4N$ | $H^+$ | 672.593 | 0.06 | | Hopmans et al. 2021 |
| $C_{9:0}$-*N*-acyl-aminotriol | São Jorge SS1 | 25.82 | $C_{44}H_{80}O_4N$ | $H^+$ | 686.608 | -0.10 | | Hopmans et al. 2021 |
| $C_{12:0}$-*N*-acyl-aminotriol | Empadadas SS7 | 29.86 | $C_{47}H_{86}O_4N$ | $H^+$ | 728.655 | 0.88 | | Hopmans et al. 2021 |
| $C_{13:0}$-*N*-acyl-aminotriol | Empadadas SS8 | 31.35 | $C_{48}H_{88}O_4N$ | $H^+$ | 742.671 | -0.59 | | Hopmans et al. 2021 |
| $C_{14:0}$-*N*-acyl-aminotriol | Cubres SS3 | 33.87 | $C_{49}H_{90}O_4N$ | $H^+$ | 756.686 | -0.17 | | Hopmans et al. 2021 |
| $C_{15:0}$-*N*-acyl-aminotriol | Cubres SS3 | 34.76 | $C_{50}H_{92}O_4N$ | $H^+$ | 770.702 | -0.25 | | Hopmans et al. 2021 |
| $C_{16:0}$-*N*-acyl-aminotriol | Cubres SS3 | 37.42 | $C_{51}H_{94}O_4N$ | $H^+$ | 784.718 | -0.24 | | Hopmans et al. 2021 |
| $C_{16:1}$-*N*-acyl-aminotriol | Cubres SS3 | 34.28 | $C_{51}H_{92}O_4N$ | $H^+$ | 782.702 | 0.31 | | Hopmans et al. 2021 |
| $C_{17:0}$-*N*-acyl-aminotriol | Lomba SS2 | 39.12 | $C_{52}H_{96}O_4N$ | $H^+$ | 798.733 | -0.58 | | Hopmans et al. 2021 |
| | | | | | | | | |
| aminotetrol | Verde SS1 | 16.26 | $C_{35}H_{64}O_4N$ | $H^+$ | 562.483 | 0.16 | | |
| | | | | | | | | |
| aminopentol | Verde SS1 | 14.69 | $C_{35}H_{64}O_5N$ | $H^+$ | 578.478 | -0.99 | | |
| unsaturated aminopentol | Verde SS1 | 13.45 | $C_{35}H_{62}O_5N$ | $H^+$ | 576.462 | 0.66 | | |
| $C_{14:0}$-*N*-acyl-aminopentol | Verde SS1 | 27.45 | $C_{49}H_{90}O_6N$ | $H^+$ | 788.676 | -0.06 | | Hopmans et al. 2021 |
| $C_{15:0}$-*N*-acyl-aminopentol | Verde SS1 | 28.35 | $C_{50}H_{92}O_6N$ | $H^+$ | 802.692 | -1.56 | | Hopmans et al. 2021 |
| $C_{16:0}$-*N*-acyl-aminopentol | Verde SS1 | 30.81 | $C_{51}H_{94}O_6N$ | $H^+$ | 816.708 | -0.77 | B6 | Hopmans et al. 2021 |
| | | | | | | | | |
| methylcarbamate-aminotriol I | Verde SS1 | 19.98 | $C_{37}H_{66}O_5N$ | $H^+$ | 604.494 | 0.02 | 3, peak e | Rush et al. 2016 |
| methylcarbamate-aminotriol II | Verde SS1 | 20.84 | $C_{37}H_{66}O_5N$ | $H^+$ | 604.494 | -0.07 | 3, peak f | Rush et al. 2016 |
| methylcarbamate-aminopentol | Negra SS2 | 16.81 | $C_{37}H_{66}O_7N$ | $H^+$ | 636.483 | 0.46 | | Rush et al. 2016 |



| | | | | | | | | |
|---|---|---|---|---|---|---|---|---|
| ethenolamine-BHT | Negra SS2 | 20.60 | $C_{37}H_{66}O_4N$ | $H^+$ | 588.499 | -0.53 | 3, peak a | Hopmans et al. 2021 |
| $\Delta^6$-ethenolamine-BHT | Verde SS1 | 18.27 | $C_{37}H_{64}O_4N$ | $H^+$ | 586.483 | -0.40 | 3, peak b and B7 | this study |
| ethenolamine-BHpentol | Negra SS2 | 18.67 | $C_{37}H_{66}O_5N$ | $H^+$ | 604.494 | -0.88 | 3, peak d | Hopmans et al. 2021 |
| ethenolamine-BHhexol | Negra SS2 | 16.64 | $C_{37}H_{66}O_6N$ | $H^+$ | 620.488 | -0.05 | 3, peak h | Hopmans et al. 2021 |
| $\Delta^{11}$-ethenolamine-BHhexol | Verde SS1 | 15.13 | $C_{37}H_{64}O_6N$ | $H^+$ | 618.473 | -1.17 | 3, peak i | this study |
| propenolamine-BHT | Negra SS2 | 21.08 | $C_{38}H_{68}O_4N$ | $H^+$ | 602.514 | -0.51 | | Hopmans et al. 2021 |
| | | | | | | | | |
| $C_{15:0}$-$N$-acyl-ethenolamine-BHhexol | Verde SS3 | 34.24 | $C_{52}H_{94}O_7N$ | $H^+$ | 844.702 | -0.37 | B8 | Hopmans et al. 2021 |
| $C_{16:0}$-$N$-acyl-ethenolamine-BHhexol | Verde SS3 | 36.76 | $C_{53}H_{96}O_7N$ | $H^+$ | 858.718 | 0.89 | | Hopmans et al. 2021 |
| $C_{17:0}$-$N$-acyl-ethenolamine-BHhexol | Verde SS3 | 37.44 | $C_{54}H_{98}O_7N$ | $H^+$ | 872.734 | 0.03 | | Hopmans et al. 2021 |
| | | | | | | | | |
| formylated aminotriol | Verde SS1 | 20.28 | $C_{36}H_{64}O_4N$ | $H^+$ | 574.483 | -0.18 | 3, peak j | Hopmans et al. 2021 |
| $\Delta^6$-formylated aminotriol | Verde SS1 | 18.03 | $C_{36}H_{62}O_4N$ | $H^+$ | 572.467 | -0.86 | 3, peak k and B9 | this study |
| formylated aminotetrol | Verde SS1 | 18.43 | $C_{36}H_{64}O_5N$ | $H^+$ | 590.478 | -1.39 | 3, peak m | Hopmans et al. 2021 |
| $\Delta^{11}$-formylated aminotetrol | Verde SS1 | 16.77 | $C_{36}H_{62}O_5N$ | $H^+$ | 588.462 | -1.00 | 3, peak n | this study |
| formylated aminopentol | Verde SS1 | 16.44 | $C_{36}H_{64}O_6N$ | $H^+$ | 606.473 | -0.70 | 3, peak o | this study |
| $\Delta^{11}$-formylated aminopentol | Verde SS1 | 14.95 | $C_{36}H_{62}O_6N$ | $H^+$ | 604.457 | -1.45 | 3, peak p | this study |
| | | | | | | | | |
| oxazinone-aminotriol | Verde SS1 | 20.72 | $C_{36}H_{62}O_4N$ | $H^+$ | 572.467 | -1.07 | 3, peak l | Elling et al. 2022 |
| | | | | | | | | |
| aminohexol I | Methylobacter-Methylotenera | 13.08 | $C_{35}H_{64}O_6N$ | $H^+$ | 594.473 | -0.36 | 5, peak a | this study |
| aminohexol II | Methylobacter-Methylotenera | 13.94 | $C_{35}H_{64}O_6N$ | $H^+$ | 594.473 | -0.08 | 5, peak b | this study |
| | | | | | | | | |
| dioxanone-methylaminotriol | Verde SS1 | 21.88 | $C_{37}H_{64}O_4N$ | $H^+$ | 586.483 | 0.72 | 3, peak c | this study |





**Table A2: Water column samples and water column measurements from the time of sampling (June 2018) discussed in this study (where D.O. = dissolved oxygen). The relative abundance of the major BHPs identified in each sample are also shown (where BHT = bacteriohopanetetrol, BHT-CE = BHT-cyclitol ether).**

| Water Column Samples | | | June 2018 Water Column | | | | | Relative Abundance of Main BHPs | | |
|---|---|---|---|---|---|---|---|---|---|---|
| Island | Lake/Lagoon | Water Depth (m) | Temp. (°C) | pH | Turbidity (NTU) | D.O. (%) | Salinity (ppt) | BHT (%) | BHT-CE (%) | Aminotriol II (%) |
| São Miguel | Azul | 1 | 20.6 | 9.9 | 11.9 | 97.9 | 0.1 | 15.8 | 60.2 | 8.7 |
| São Miguel | Azul | 5 | 19.9 | 8.4 | 14.1 | 84.1 | 0.1 | 23.0 | 54.8 | 9.0 |
| São Miguel | Azul | 15 | 14.2 | 7.8 | 8.7 | 63.3 | 0.1 | 22.2 | 27.4 | 23.0 |
| Sao Miguel | Empadadas Norte | 1 | 16.5 | 8.9 | 20.9 | 93.0 | 0.0 | 23.7 | 4.8 | 17.8 |
| São Jorge | Cubres East | 0 | 22.6 | 9.9 | 6.8 | 95.0 | 9.7 | 54.1 | 0.0 | 20.9 |
| São Jorge | Cubres West | 0.5 | 24.1 | 9.9 | 4.9 | - | 23.8 | 47.5 | 0.0 | 34.6 |
| Flores | Funda | 1 | 19.7 | 11.3 | 29.4 | 96.4 | 0.1 | 29.0 | 53.9 | 1.7 |
| Flores | Funda | 6 | 15.2 | 8.2 | 5.4 | 57.3 | 0.1 | 23.2 | 57.6 | 5.1 |
| Flores | Funda | 29 | 12.3 | 8.3 | 4.0 | 24.1 | 0.1 | 6.4 | 6.8 | 43.5 |
| Flores | Negra | 1 | 18.7 | 11.3 | 57.4 | 100.6 | 0.1 | 8.0 | 90.8 | 0.4 |
| Flores | Negra | 7 | 14.7 | 9.7 | 7.4 | 77.6 | 0.1 | 13.4 | 84.5 | 0.5 |
| Flores | Negra* | 100 | - | - | - | - | - | 22.7 | 38.5 | 20.3 |

*Note: Measurements in Lake Negra are only made up to ~50 m water column depth.





**Table A3: Location and characteristics of the sediment sample sites discussed in this study (where Rep. = replicates, TN = total nitrogen, and TOC = total organic carbon). The relative abundance of the major BHPs identified in each sample are also shown (where Adeno. = adenosylhopane, N1-methylino. = N1-methylinosylhopane, BHT = bacteriohopanetetrol, and BHT-CE = BHT-cyclitol ether).**

| | Site | Sediment samples | | | | Relative Abundance of Main BHPs | | | | | | |
|---|---|---|---|---|---|---|---|---|---|---|---|---|
| Island | Lake/ Lagoon | Depth (m) | Rep. | TN (%) | TOC (%) | Adeno. (%) | N1-methylino. (%) | BHT (%) | BHT-CE (%) | Aminotriol II (%) | Aminopentol (%) | Ethenolamine-BHT (%) |
| São Miguel | Azul | 5 | 2 | 0.59 | 5.14 | 3.1 | 1.2 | 9.0 | 10.1 | 17.2 | 28.7 | 2.1 |
| São Miguel | Azul | 23 | 0 | 0.60 | 4.43 | 6.3 | 2.9 | 13.0 | 9.2 | 13.8 | 16.1 | 2.4 |
| São Miguel | Verde | 5 | 0 | 1.22 | 8.81 | 1.4 | 1.0 | 12.5 | 11.2 | 16.3 | 11.7 | 4.0 |
| São Miguel | Verde | 22.5 | 0 | 0.86 | 8.18 | 1.7 | 2.0 | 8.2 | 19.1 | 13.8 | 22.2 | 2.0 |
| Sao Miguel | Empadadas Norte | 2 | 4 | 0.82 | 8.26 | 10.4 | 13.2 | 9.5 | 7.4 | 11.4 | 8.7 | 1.7 |
| São Jorge | São Jorge | 2.5 | 2 | 0.97 | 11.00 | 13.8 | 15.4 | 8.9 | 8.0 | 7.6 | 1.3 | 1.8 |
| São Jorge | Cubres East | 2 | 0 | 1.95 | 13.79 | 1.9 | 1.1 | 28.0 | 0.0 | 11.3 | 0.2 | 11.4 |
| São Jorge | Cubres West | 2 | 0 | 1.92 | 13.89 | 3.6 | 1.4 | 23.6 | 4.0 | 19.0 | 2.3 | 8.1 |
| Flores | Funda | 29.7 | 0 | 0.89 | 7.18 | 2.2 | 1.1 | 13.8 | 19.4 | 15.5 | 11.2 | 4.6 |
| Flores | Lomba | 14.3 | 2 | 1.94 | 19.45 | 11.6 | 7.6 | 10.0 | 9.6 | 12.4 | 8.1 | 2.3 |
| Flores | Negra | 114 | 2 | 0.83 | 6.77 | 5.6 | 0.8 | 11.6 | 19.7 | 11.2 | 5.5 | 3.1 |



**Table A4:** *Methylobacter-Methylotenera* **co-culture (n=3) BHP distributions with BHPs reported as relative abundance and the standard deviation between culture samples. The retention times (tr) and calculated exact mass (M$_{calc}$) are also shown.**

| Bacteriohopanepolyol | tr (min) | M$_{calc}$ ($m/z$) | Relative Abundance (%) | Standard deviation |
|---|---|---|---|---|
| adenosylhopane | 21.77 | 662.50 | 0.174 | 0.006 |
| 2Me-adenosylhopane$_{HG-diMe}$ | 25.25 | 704.547 | 0.002 | 0.001 |
| aminotriol II | 17.52 | 546.49 | 43.308 | 1.027 |
| $\Delta^6$-aminotriol | 15.47 | 544.47 | 0.067 | 0.032 |
| $\Delta^{11}$- aminotriol | 15.88 | 544.47 | 3.454 | 0.133 |
| C$_{14:0}$-$N$-acyl-aminotriol | 33.91 | 756.69 | 0.006 | 0.002 |
| C$_{16:0}$-$N$-acyl-aminotriol | 37.41 | 784.72 | 0.007 | 0.004 |
| C$_{16:1}$-$N$-acyl-aminotriol | 34.44 | 782.70 | 0.079 | 0.043 |
| aminotetrol | 16.30 | 562.48 | 6.381 | 0.188 |
| aminopentol | 14.69 | 578.48 | 43.441 | 1.242 |
| unsaturated aminopentol | 13.51 | 576.46 | 2.552 | 0.117 |
| C$_{14:0}$-$N$-acyl-aminopentol | 27.49 | 788.68 | 0.008 | 0.001 |
| C$_{15:0}$-$N$-acyl-aminopentol | 30.82 | 816.71 | 0.005 | 0.001 |
| methylcarbamate-aminotriol II | 20.89 | 604.49 | 0.126 | 0.030 |
| methylcarbamate-aminotetrol | 18.96 | 620.49 | 0.003 | 0.003 |
| methylcarbamate-aminopentol | 16.87 | 636.48 | 0.024 | 0.004 |
| ethenolamine-BHT | 20.67 | 588.50 | 0.039 | 0.013 |
| ethenolamine-BHhexol | 16.70 | 620.49 | 0.009 | 0.009 |
| formylated-aminotriol | 20.28 | 574.48 | 0.033 | 0.012 |
| formylated-aminopentol | 16.44 | 606.47 | 0.019 | 0.012 |
| aminohexol I | 13.08 | 594.47 | 0.131 | 0.018 |
| aminohexol II | 13.94 | 594.47 | 0.117 | 0.016 |






**Appendix B. Identification of BHPs**



| BHP | R₁ | R₂ | R₃ | R₄ | R₅ | R₆ | R₇ | R₈ |
|---|---|---|---|---|---|---|---|---|
| **BHT** | H | H | H | H | OH | OH | OH | OH |
| **2Me-BHT** | CH₃ | H | H | H | OH | OH | OH | OH |
| **3Me-BHT** | H | CH₃ | H | H | OH | OH | OH | OH |
| **BHpentol** | H | H | H | OH | OH | OH | OH | OH |
| **BHhexol** | H | H | OH | OH | OH | OH | OH | OH |
| **aminotriol** | H | H | H | H | OH | OH | OH | NH₂ |
| **2Me-aminotriol** | CH₃ | H | H | H | OH | OH | OH | NH₂ |
| **Me-aminotriol*** | H | H | H | H | OH | OH | OH | NH₂ |
| **3Me-aminotriol** | H | CH₃ | H | H | OH | OH | OH | NH₂ |
| **aminotetrol** | H | H | H | OH | OH | OH | OH | NH₂ |
| **aminopentol** | H | H | OH | OH | OH | OH | OH | NH₂ |

*Note: The extra methyl group does not occur at the C-2 or C-3 position. The position is unknown.

**Figure B1: Core structure of BHPs discussed in the text with the carbon positions and the rings labelled. More complex structures are shown in the figures.**




**B1 Novel nucleoside BHPs**

We identified nucleoside BHPs based on Hopmans et al. (2021) in surface sediments and water column samples from the Azores Archipelago. Fig. B2 shows the partial mass chromatograms of adenosyl-containing nucleoside BHPs with increasing degrees of methylation, found in surface sediment samples from São Jorge and Empadadas. In addition to the adenosyl-BHPs previously identified by Hopmans et al. (2021; peaks a, c-e, h-j, and m-o), we observe several additional peaks (b*, f*, g*, k*, and l*). In the mass chromatogram $m/z$ 676.516, an early eluting peak (b*) occurs in both samples. The MS$^2$ spectrum for peak b* contains one fragment ion related to the head group at $m/z$ 150.077 ($C_6H_8N_5^+$, $\Delta$ ppm -0.52), which suggests that the adenosyl head group is methylated. Therefore, we tentatively assign peak b* as early eluting adenosylhopane$_{HG-Me}$ (assigned elemental composition (AEC) = $C_{41}H_{66}O_3N_5$, $m/z$ 676.516, $\Delta$ ppm 0.14; Fig. B2d). The mass chromatogram of $m/z$ 690.532, contains two early eluting peaks in São Jorge (peaks f* and g*) and one early eluting peak in Empadadas (peak g*). Peaks f* and g* are putatively identified as Me-adenosylhopane$_{HG-Me}$ and adenosylhopane$_{HG-diMe}$ based on the dominant fragment ions related to the head group at $m/z$ 150.077 ($C_6H_8N_5^+$, $\Delta$ ppm -0.52; Fig. B2e) and $m/z$ 164.093 ($C_7H_{10}N_5^+$, $\Delta$ ppm -1.09; Fig. B2f), respectively. In the sediment samples from Lake São Jorge, the mass chromatogram for $m/z$ 704.547 contains one early eluting peak (peak k*). The MS$^2$ spectrum of peak k* is characterized by a fragment at $m/z$ 164.093 ($C_7H_{10}N_5^+$, $\Delta$ ppm -0.78; Fig. B2g), which is associated with a di-methylated head group. Thus, we tentatively identify peak k* as Me-adenosylhopane$_{HG-diMe}$. In Empadadas, we observe another early eluting peak (peak l*) in the mass chromatogram for $m/z$ 704.547 with an MS$^2$ spectrum that contains two main mass fragments associated with the adenosyl head group: $m/z$ 150.077 ($C_6H_8N_5^+$, $\Delta$ ppm -0.85) and $m/z$ 164.093 ($C_7H_{10}N_5^+$, $\Delta$ ppm -1.27; Fig. B2h). Peak l* is likely another isomer of diMe-adenosylhopane$_{HG-Me}$, but is co-eluting with Me-adenosylhopane$_{HG-Me}$ (peak i) as suggested by the fragment $m/z$ 150.077 ($C_6H_8N_5^+$).

Inosylhopanes, described in Hopmans et al. (2021), were identified in surface sediments from Lake Empadadas, (peaks a, d, and e; Fig. B3a) as well as several novel isomers of methylated-inosylhopanes (peaks b*, c*, f*, and g*; Fig. B3a). The mass chromatogram of $m/z$ 691.516 contains two early eluting peaks (peak b* and c*). Both mass spectra contain one primary fragment associated with the head group, $m/z$ 137.046 ($C_5H_5ON_4^+$, peak b*, $\Delta$ ppm -1.11 and peak c*, $\Delta$ ppm -0.82; Fig. B3b), indicating an inosine head group. Thus, we identify these peaks as Me-inosylhopane with the methylation occurring on the core structure. The retention time difference between inosylhopane and peaks b* (0.12 min) and c* (0.57 min), is similar to retention time difference between BHT and 2MeBHT (0.14 min) and MeBHT with a methylation at an unknown position in the ring system (0.58 min), respectively (Hopmans et al., 2021). Therefore, we tentatively assign peak b* as 2Me-inosylhopane and peak c* as Me-inosylhopane, unknown isomer. For peak f* we observe a fragment of $m/z$ 151.061 ($C_6H_7ON_4^+$, $\Delta$ ppm -1.14; Fig. B3c), suggesting a methylation on the inosine head group. We propose that peak f* is an isomer of methyl-N1-methylinosylhopane (elemental composition (EC) = $C_{42}H_{67}O_4N_4$, $m/z$ 691.516). Based on the fragment $m/z$ 151.061 ($C_6H_7ON_4^+$, $\Delta$ ppm -1.08; Fig. B3d), peak g* is another isomer of Me-N1-methylinosylhopane with unknown positions of the methyl groups on the ring system and head group. However, we also observe an $m/z$ 164.093 ($C_7H_{10}N_5^+$, $\Delta$ ppm -1.15) fragment in the MS$^2$ spectra of peak g*. We attribute this to potential a potential co-eluting peak that we are unable distinguish.



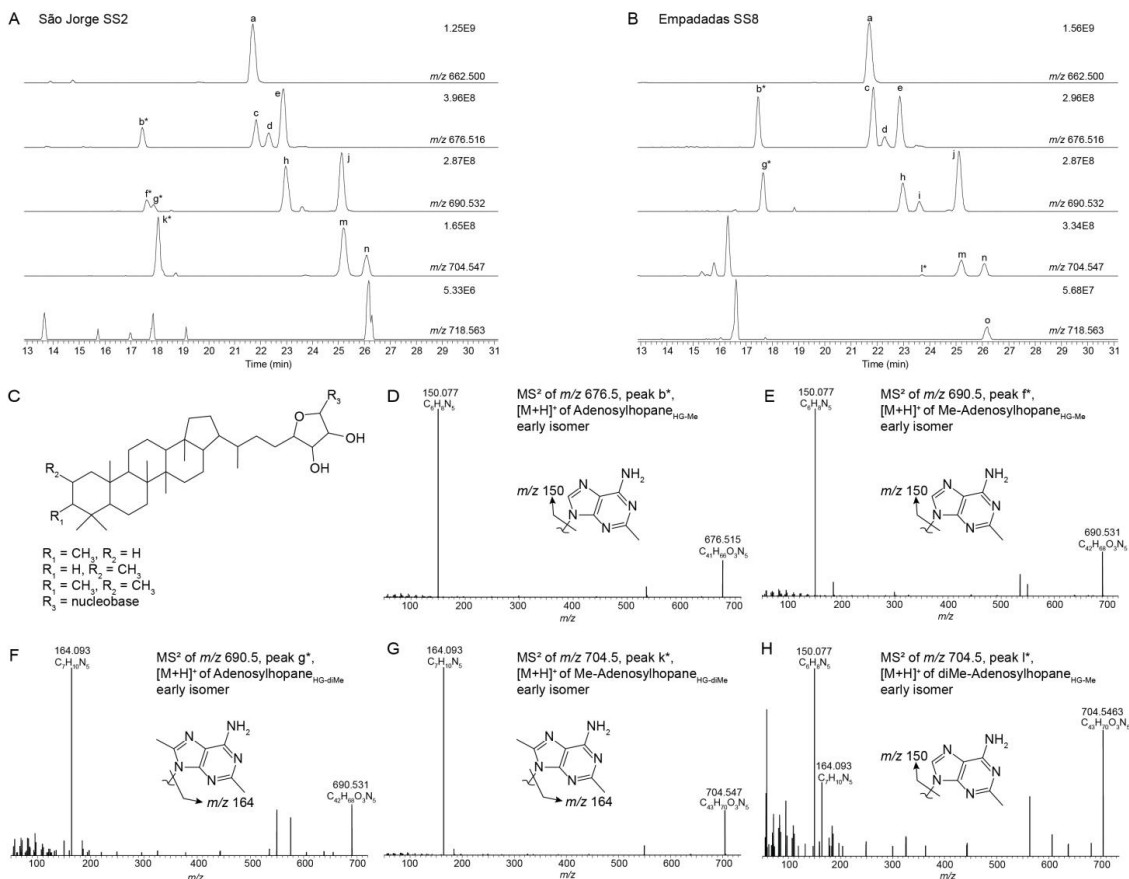

**Figure B2: Partial mass chromatograms of adenosyl-containing nucleoside BHPs in a sediment sample from (A) Lake São Jorge and (B) Lake Empadadas showing the exact mass and intensity of the highest peak in arbitrary units (AU). Peaks identified as novel nucleoside BHP isomers are marked with a '*'. (C) general structure of a nucleoside BHP (D-H) MS$^2$ of novel early-eluting nucleoside BHP isomers identified in Azorean lakes and a proposed structure for each nucleobase. Note, the mass spectra of peak l* (panel H) appears to be a mixed spectrum of co-eluting nucleoside BHPs.**



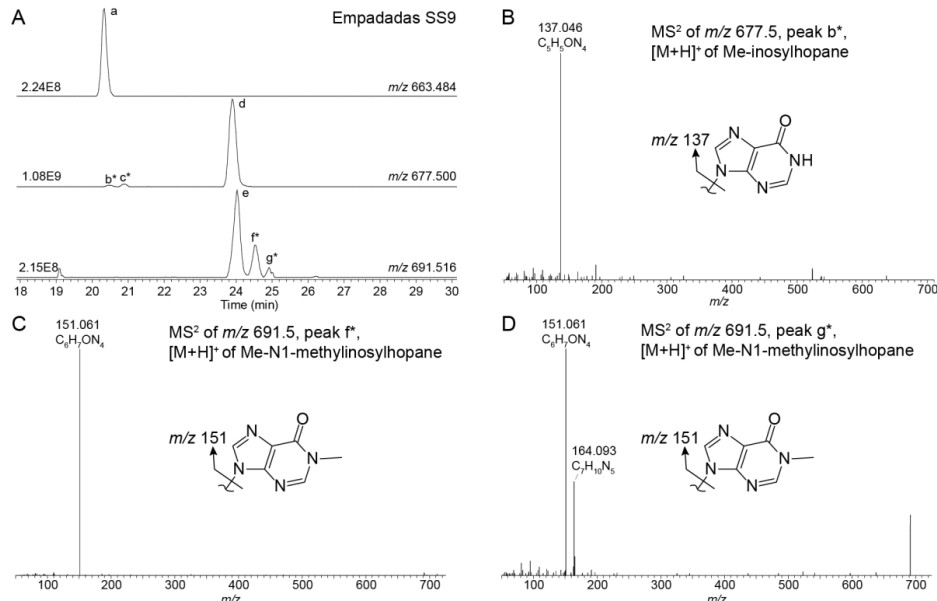

**Figure B3: (A)** Partial mass chromatogram of inosylhopane (peak a), N1-methylinosylhopane (peak d), and 2-methyl-N1-methylinosylhopane (peak e) in a sediment sample from Lake Empadadas. **(B)** A MS$^2$ of newly identified peak of putative Me-inosylhopane (peak b*) with the proposed structure for the nucleobase shown. MS$^2$ of newly identified 2-methyl-N1-methylinosylhopane isomers for **(C)** peak f* and **(D)** peak g* with the structures of the proposed nucleobases. Note, the mass spectra for peak g* appears to be a mixed spectrum.




**B2 Methoxylated-bacteriohopanetetrol**

In the mass chromatogram of *m/z* 578.514 ($C_{36}H_{68}O_4N^+$) we identified two peaks that elute after the expected retention times of 2MeBHT (peak a) and 3MeBHT (Fig. B4a). The MS[2] spectrum of the peak b at 21.71 mins (Fig. B4b) shows a distinct loss of three hydroxyl moieties: *m/z* 543.477 ($C_{36}H_{63}O_3^+$, Δ ppm -0.68), *m/z* 525.467

($C_{36}H_{61}O_2^+$, Δ ppm -0.48), and *m/z* 507.459 ($C_{36}H_{59}O^+$, Δ ppm -5.37) and a loss of 32 Da ($CH_3OH$) from *m/z* 525.466 ($C_{36}H_{61}O_2^+$, Δ ppm -0.48) to *m/z* 493.441 ($C_{35}H_{57}O^+$, Δ ppm -0.26). The lower mass range of the mass spectrum is comparable to that of BHT (Hopmans et al., 2021). We therefore tentatively identify this BHP as methoxy-BHT. Peak c at 22.30 mins shows a similar loss of three hydroxyl moieties and a loss of 32 Da. The later elution time of this peak could result from a different position of the methoxy-group on the side-chain. The

position of the methoxy group in both isomers is unknown. We tentatively identify these peaks as methoxylated-BHTs.

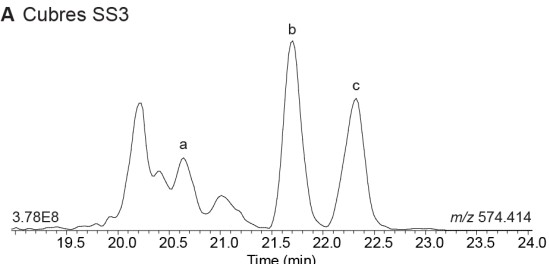
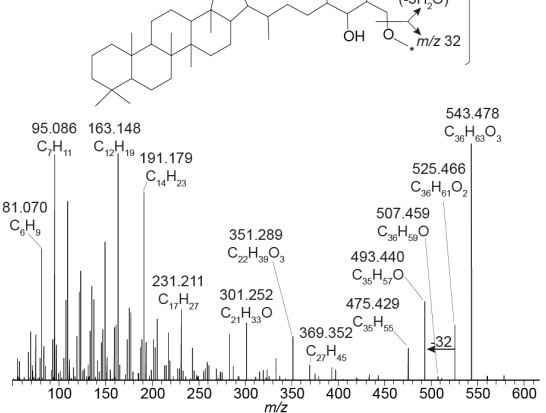

**Figure B4: (A) Partial mass chromatogram of 2MeBHT (peak a), methoxylated-BHT I (peak b), and methoxylated-BHT II in a sediment sample from Cubres East. (B) MS² of the methoxylated-BHT I (peak b) and the proposed structure for methoxylated BHT. The asterisk (*) indicates that the position of the methoxy moiety is unknown.**



**B3 Bacteriohopanetetrol-methyl-cyclitol ether**

In the mass chromatogram of $m/z$ 722.555, the calculated exact mass of MeBHT-cyclitol ether (CE), from Verde

SS1 we observe two peaks (Fig. B5a): a peak at 17.25 mins (peak b) and 3MeBHT-CE ($C_{42}H_{76}O_8N^+$; peak c) at 18.63 mins. The $MS^2$ spectrum of peak b (Fig. B5b) shows the consecutive losses of two hydroxyl moieties producing fragment ions at $m/z$ 704.543 ($C_{42}H_{74}O_7N^+$, $\Delta$ ppm 3.85) and $m/z$ 686.529 ($C_{42}H_{72}O_6N^+$, $\Delta$ ppm 2.95), however, the expected loss of a third hydroxyl moiety ($m/z$ 668.520, $C_{42}H_{70}O_5N^+$) is not observed, likely due to the low intensity. In the lower mass range, the fragments follow the fragmentation pattern for a cyclitol ether head

group described in Hopmans et al. (2021), but with an additional mass of 14 Da. The intact head group corresponds to the fragment $m/z$ 194.102 ($C_7H_{16}O_5N^+$, $\Delta$ ppm -0.31) followed by the loss of several hydroxyl moieties: $m/z$ 176.091 ($C_7H_{14}O_4N^+$, $\Delta$ ppm 1.05), $m/z$ 158.081 ($C_7H_{12}O_3N^+$, $\Delta$ ppm -2.47), $m/z$ 140.070 ($C_7H_{10}O_2N^+$, $\Delta$ ppm 0.32). We also observe two fragments that correspond to the intact polar head group with two additional carbon atoms, originating from the side chain after fragmentation at the C-33 and C-34 bond: $m/z$ 236.113 ($C_9H_{18}O_6N^+$,

$\Delta$ ppm 0.06) and $m/z$ 218.102 ($C_9H_{16}O_5N^+$, $\Delta$ ppm 0.09). Based on the $MS^2$ spectrum, we tentatively identify this peak as a BHT cyclitol ether with a methylation on the head group: BHT-Me-CE (AEC = $C_{42}H_{76}O_8N^+$, $m/z$ 722.557, $\Delta$ ppm -1.64). Note, that we do not observe a loss of 32 Da indicative of a methoxy moiety ($CH_4O^+$), therefore we propose that the methyl-group occurs on the ring structure of the cyclitol ether.

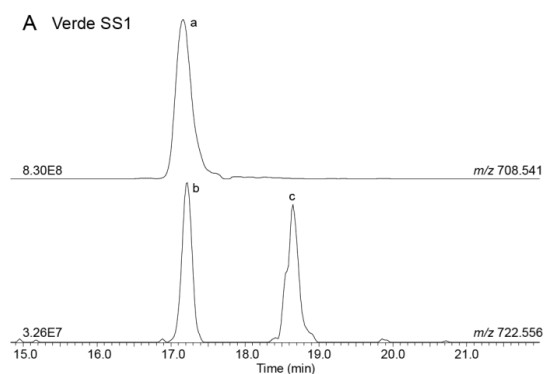

**Figure B5: (A) Partial mass chromatogram of BHT cyclitol ether (BHT-CE, peak a) and newly proposed BHT-Me-CE (peak b) in a sediment sample from Lake Verde. (B) MS² spectrum of [M+H]⁺ BHT-Me-CE (peak b) and a proposed structure for BHT-Me-CE with the diagnostic fragmentation of the cyclitol ether indicated (Note: the asterisk (*) indicates that the position of the methyl group is unknown, the additional methylation could also occur at the C-39 or C-40 position).**

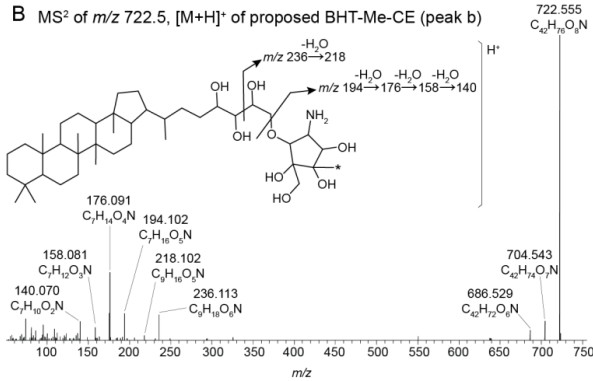





## B4 Ethenolamine BHPs

In our samples we identified ethenolamine-BHT (peak a), ethenolamine-BHpentol (peak d), and ethenolamine-BHhexol (peak h) after Hopmans et al. (2021; Fig. 3a). In addition, we identified unsaturated ethenolamine-BHT (peak b) and unsaturated ethenolamine-BHhexol (peak i) in the mass chromatograms of $m/z$ 586.483 ($C_{37}H_{64}O_4N^+$) and $m/z$ 618.474 ($C_{37}H_{64}O_6N^+$), respectively. The fragmentation spectra of both unsaturated ethenolamine-BHPs

(Fig. 3e and B6, respectively) showed the characteristic loss of 41 Da ($C_2H_3N$) and 59 Da ($C_2H_5ON$) related to the loss of the ethenolamine moiety. In both fragmentation spectra we observe the expected fragment for an unsaturation on the BHP core structure, i.e. $m/z$ 473.413 ($C_{35}H_{53}^+$, Δ ppm 1.54) for an unsaturated BHT and $m/z$ 469.382 ($C_{35}H_{49}^+$, Δ ppm 2.08) for an unsaturated BHhexol. The offset in retention time between ethenolamine-BHT and unsaturated-ethenolamine-BHT (2.33 mins) and ethanolamine-BHhexol and unsaturated-ethenolamine-

BHhexol (1.51 mins), respectively, is similar to the offset between BHT and $Δ^6$-BHT (2.34 mins) and BHT and $Δ^{11}$-BHT (1.38 mins), respectively, observed by Hopmans et al. (2021). Based on the retention time offset, we propose that the unsaturation in peak b and i are at the $Δ^6$- and $Δ^{11}$-position, respectively. In the lower mass range of the MS$^2$ spectrum of the proposed $Δ^6$-ethenolamine-BHT we observe some deviations from the expected fragments for a $Δ^6$-BHT, i.e. a relatively dominant $m/z$ 177 and $m/z$ 201 instead of $m/z$ 203, which is expected for

a $Δ^6$-BHT core (Hopmans et al., 2021). Perhaps the presence of a conjugation alters the fragmentation pattern of an unsaturated core to a small degree. The MS$^2$ spectrum of the proposed $Δ^{11}$-ethenolamine BHhexol (peak i) also shows the dominant fragment ion at $m/z$ 177, but does show the expected fragment ion at $m/z$ 203 for a $Δ^{11}$ unsaturation.

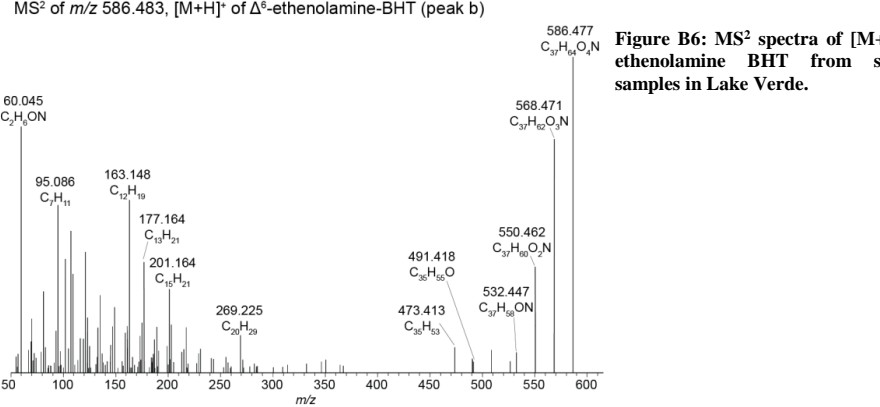

MS$^2$ of $m/z$ 586.483, [M+H]$^+$ of $Δ^6$-ethenolamine-BHT (peak b)

**Figure B6: MS$^2$ spectra of [M+H]$^+$ $Δ^6$-ethenolamine BHT from sediment samples in Lake Verde.**

## B5 Formylated-aminoBHPs


We detected two unknown composite BHPs previously described in Hopmans et al. (2021) in the partial mass chromatograms of $m/z$ 574.483 ($C_{36}H_{64}O_4N^+$) and $m/z$ 590.483 ($C_{36}H_{64}O_5N^+$). In addition to several consecutive losses of water representing 3 or 4 hydroxyl moieties on the side chain, the fragmentation spectra of both of these unknown composite BHPs is characterized by an initial loss 28 Da (CO). Here, we detected these novel BHPs in


sediment from Lake Verde (Fig. 3b, peaks j and m). In addition, we identified an additional BHP with $m/z$ 606.473 and an AEC of $C_{36}H_{64}O_6N^+$ (Fig. 3b and f, peak o) with a similar fragmentation: a loss of 28 Da (CO) from the



original mass fragment produces the fragment at *m/z* 578.484 ($C_{35}H_{64}O_5N^+$, $\Delta$ ppm -5.81) and a loss of a hydroxyl moiety (*m/z* 588.461, $C_{36}H_{62}O_5N^+$, $\Delta$ ppm 1.29) followed by a loss of 28 Da (CO) results in the fragment at *m/z* 560.468 ($C_{35}H_{62}O_4N^+$, $\Delta$ ppm -1.76). We also observe five sequential losses of hydroxyl moieties from the parent

ion to produce *m/z* 588.461 ($C_{36}H_{62}O_5N^+$, $\Delta$ ppm 2.96), *m/z* 570.451 ($C_{36}H_{60}O_4N^+$, $\Delta$ ppm -0.17), *m/z* 552.441 ($C_{36}H_{58}O_3N^+$, $\Delta$ ppm 2.03), *m/z* 534.431 ($C_{36}H_{56}O_2N^+$, $\Delta$ ppm -1.92), and *m/z* 516.420 ($C_{36}H_{54}ON^+$, $\Delta$ ppm 1.09), respectively. In the fragmentation spectra of peaks j, m, and o we observe losses of 45 Da ($CONH_3$), e.g. from *m/z* 534 to *m/z* 489 for peak o (Fig. 3f). Based on these losses we propose that these novel composite BHPs are a series of formylated-aminoBHPs: formylated-aminotriol (peak j), formylated-tetrol (peak m) and formylated-

aminopentol (peak o). The proposed structure for formylated-aminopentol is shown in Fig. 3f with the diagnostic fragmentations indicated.

We also observe unsaturated versions of these formylated-aminotriol, formylated-aminotetrol, and formylated-aminopentol based on the elemental composition $C_{36}H_{62}O_4N^+$ (*m/z* 572.467, peak k), $C_{36}H_{62}O_5N^+$ (*m/z* 588.462,

peak n), $C_{36}H_{62}O_6N^+$ (*m/z* 604.458, peak p), respectively. In the lower mass range of the *m/z* 572.467 spectrum, we observe dominant *m/z* 163.148 ($C_{12}H_{19}^+$, $\Delta$ ppm -0.14), *m/z* 177.163 ($C_{13}H_{21}^+$, $\Delta$ ppm 0.10), *m/z* 201.164 ($C_{15}H_{21}^+$, $\Delta$ ppm 0.53), and *m/z* 203.179 ($C_{15}H_{23}^+$, $\Delta$ ppm -0.46) fragments and a notable absence of a dominant *m/z* 191.179 ($C_{14}H_{23}^+$). This is similar to the fragmentation patterns observed for unsaturated ethenolamine-BHT (Fig. B7) and unsaturated ethenolamine-BHhexol (Fig. 3e) as discussed in section B4. We also observe the

corresponding fragments of the D and E ring with a partially de-hydroxylated side-chain in the spectra: *m/z* 350.271 ($C_{21}H_{36}O_3N^+$, $\Delta$ ppm -3.31) and after two additional hydroxyl losses, *m/z* 314.249 ($C_{21}H_{32}ON^+$, $\Delta$ ppm -5.95), and the loss of the hexane ring on the side-chain to get *m/z* 231.210 ($C_{17}H_{27}^+$, $\Delta$ ppm 0.03). This indicates that the unsaturation is not on the side-chain or the D and E rings. Based on the presence of the *m/z* 201.164 fragment and a similar offset in retention times between formylated-aminotriol and unsaturated formylated-

aminotriol (2.25 mins; Table A1) to that of ethenolamine-BHT and unsaturated ethenolamine-BHT (2.33 mins; Table A1), we tentatively assign the double bond to the $\Delta^6$ position. In *m/z* 588.462 ($C_{36}H_{62}O_5N^+$; Fig. 3b), we identify peak n as unsaturated formylated-aminotetrol based on the $MS^1$ spectrum. However, due to a low $MS^2$ spectrum we cannot identify the position of the double bond. Based on the retention time, the double bond is likely at the $\Delta^6$ position. The difference in retention time between formylated-aminopentol and unsaturated formylated-

aminopentol (1.49 mins) is similar to that of the ethenolamine-BHhexol and unsaturated ethenolamine-BHhexol

MS² of *m/z* 572.4, [M+H]⁺ of $\Delta^6$-formylated-aminotriol (peak k)

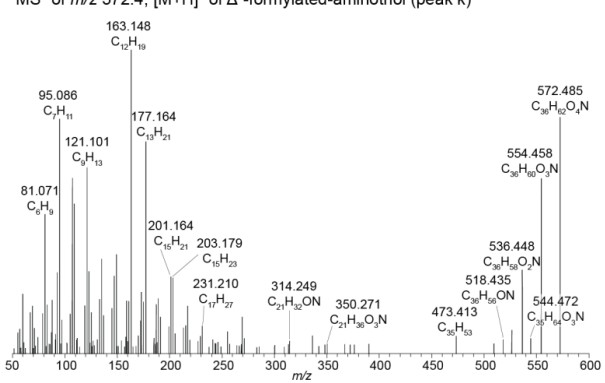

**Figure B7: MS² spectra of [M+H]⁺ $\Delta^6$-formylated aminotriol from sediment samples in Lake Verde.**



(1.51 mins), therefore the double bond likely occurs in the same place for both compounds. We therefore tentatively identify peak p as $\Delta^{11}$-formylated aminopentol.

### B6 Oxazinone-aminotriol

In the mass chromatogram for $m/z$ 572.467, in addition to the unsaturated formylated-aminotriol (peak k), we
observe a later-eluting peak ($C_{36}H_{62}O_4N^+$; Fig. 3b and d, peak l). The MS$^2$ spectrum of this peak contains a loss of 44 Da ($CO_2$) from the original mass fragment to produce $m/z$ 528.477 ($C_{35}H_{62}O_2N^+$, $\Delta$ ppm -1.33) followed by a loss of two hydroxyl moieties ($m/z$ 510.466, $C_{35}H_{60}ON^+$, $\Delta$ ppm 1.36 and $m/z$ 492.457, $C_{35}H_{58}N^+$, $\Delta$ ppm -2.92). We propose that this is the same compound identified in Elling et al. (2022). The authors describe an acetylated compound ($m/z$ 656.495; $C_{40}H_{66}O_6N^+$), with the unique loss of 44 Da ($CO_2$), the loss of two hydroxyl moieties,
and an additional loss of 17 Da ($NH_3^+$). We indicate the loss of 17 Da ($NH_3^+$) in the proposed structure; however, we do not observe this loss in our MS$^2$ spectrum, which could be attributed to low peak intensity. Based on the proposed structure, this appears to be the cyclized form of the formylated-aminotriol. The IUPAC name for the cyclized part of the BHP is 1,3-oxazinan-2-one (National Center for Biotechnology Information, 2023a), therefore we refer to this compound as oxazinone-aminotriol.

### B7 Dioxanone-methylaminotriol

In the mass chromatogram of $m/z$ 586.483 ($C_{37}H_{64}O_4N^+$; $\Delta$ ppm 0.72), an additional peak occurs at 21.88 mins (Fig. 3a, peak c). Similar to the above described formylated-aminoBHPs, the MS$^2$ spectra is characterized by the distinct loss of 28 Da producing $m/z$ 558.488 ($C_{36}H_{64}O_3N^+$; $\Delta$ ppm -0.37; Fig. 3c). In addition, a loss of 31 Da ($CNH_5$) is observed from $m/z$ 522.467 ($C_{36}H_{60}ON^+$; $\Delta$ ppm 0.04) to produce the fragment at $m/z$ 491.424
($C_{35}H_{55}O^+$; $\Delta$ ppm 1.98), indicating a second conjugated moiety. Based on the elemental composition there appears to be an additional DBE (double bond equivalent) in the structure, but there is no indication the unsaturation is located in the core structure as the lower mass range of the fragmentation spectrum is similar to what is expected for BHT. We therefore assume one of the functionalities is part of a cyclized structure and propose this BHP is a type of formylated aminotriol, where the formic acid is part of a cyclic structure and the
terminal amino group is methylated. The IUPAC name for the cyclic part of the structure is 1,3-dioxan-2-one (National Center for Biotechnology Information, 2023b), therefore we refer to this compound as a dioxanone-methylaminotriol. The proposed structure with key fragmentations is shown in Fig. 3c.

### B8 Aminohexol BHPs

A search for additional BHPs using the $m/z$ 191.179 trace ($C_{14}H_{22}^+$) revealed two early eluting peaks in the mass
chromatogram for $m/z$ 594.473 ($C_{35}H_{64}O_6N^+$) in the *Methylobacter-Methylotenera* co-culture (Fig. 9a). The MS$^2$ spectrum for aminohexol I (Fig. 9b) shows the consecutive loss of five hydroxyl moieties to produce fragments at $m/z$ 594.473 ($C_{35}H_{64}O_6N^+$, $\Delta$ ppm -0.36), $m/z$ 576.462 ($C_{35}H_{62}O_5N^+$, $\Delta$ ppm 0.59), $m/z$ 558.454 ($C_{35}H_{60}O_4N^+$, $\Delta$ ppm -3.55), $m/z$ 540.441 ($C_{35}H_{58}O_3N^+$, $\Delta$ ppm 0.83), $m/z$ 522.432 ($C_{35}H_{56}O_2N^+$, $\Delta$ ppm -2.30), and $m/z$ 504.418 ($C_{35}H_{54}ON^+$, $\Delta$ ppm 3.04). Based on the presence of $m/z$ 369.354 ($C_{27}H_{45}^+$, $\Delta$ ppm -6.91) in the MS$^2$ spectrum, the
additional hydroxyl moiety does not occur on the ring system. The position of the additional hydroxyl moiety on the tail is unknown. Similarly, the MS$^2$ spectrum for aminohexol II (Fig. 9c) also shows a loss of five hydroxyl moieties, and an additional loss of 29 Da ($CH_3N$) from $m/z$ 504.422 ($C_{35}H_{54}ON^+$, $\Delta$ ppm -3.01) to $m/z$ 475.393





($C_{34}H_{51}O^+$, $\Delta$ ppm 0.99). However, in the fragmentation spectrum of peak b, we observe several fragments that appear to represent the functionalized tail at *m/z* 180.086 ($C_6H_{14}O_5N^+$, $\Delta$ ppm -0.12), *m/z* 162.076 ($C_6H_{12}O_4N^+$, $\Delta$

ppm 0.95), *m/z* 144.0655 ($C_6H_{10}O_3N^+$, $\Delta$ ppm -0.35), and *m/z* 126.055 ($C_6H_8O_2N^+$, $\Delta$ ppm -1.55). The complementary ion to the tail fragments is observed at *m/z* 395.368 ($C_{29}H_{47}^+$, $\Delta$ ppm -0.06), indicating fragmentation at the C22-C30 bond. The presence of the *m/z* 369.354 ($C_{27}H_{45}^+$, $\Delta$ ppm -0.20) fragment indicates a hopanoid core without the addition of an extra hydroxyl group. We propose that in this isomer, the additional hydroxyl group is located at C-22 on the side-chain, resulting in the initial fragmentation of the side-chain and the

progressive loss of the hydroxyl groups from the side-chain fragment. Based on the MS$^2$ spectra, we tentatively identify these compounds as aminohexol BHPs.





**Appendix C Additional figures**

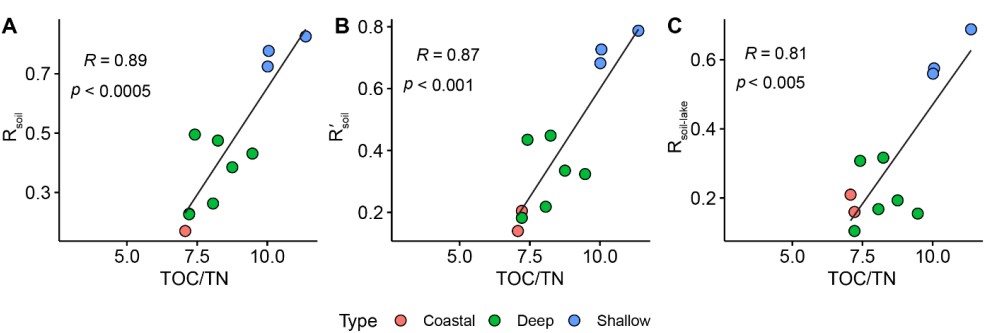


**Figure C1: Correlation between (A) R$_{soil}$, (B) R$_{soil'}$, and (C) R$_{soil-lake}$ with TOC/TN. The different colors represent the different types of samples sites in this study: coastal (Cubres East and West), deep (Azul, Funda, and Negra), and shallow (Empadadas, São Jorge, and Lomba).**




**Data availability**

Mass spectra published in this manuscript will be added to the Global Natural Products Social (GNPS) Molecular

Networking database (https://gnps.ucsd.edu/ProteoSAFe/static/gnps-splash.jsp). All additional data produced in

this study will be published in the supplementary material.

**Author contribution**

The study was conceptualized by NR, LAA-Z, and DR. The investigation was carried out by all co-authors. Formal

analysis was conducted by NR, ECH, and DM. ECH, PMR, VG, ACC, and LV contributed resources to make this

project possible. Funding was acquired by LAA-Z, PMR, LV, and DR. NR wrote the manuscript, and all co-

authors contributed to the manuscript.

**Competing Interests**

The authors declare that they have no conflict of interest.

**Acknowledgements**

This work was supported by the Luso-American Foundation ("Crossing the Atlantic") (LAA-Z), the National

Funds through FCT - Foundation for Science and Technology under the project UIDB/50027/2020, PMR by

(DL57/2016/ ICETA/EEC2018/25) grant, and the NWO gravitation grant for Soehngen Institute of Anaerobic

Microbiology (024.002.002). We would like to thank everyone who participated and assisted with field campaign

to the Azores Archipelago in 2018, particularly E. Zettler. Further, we would like to thank Dr. S. van Grinsven

for the co-culture and advice. We thank Dr. N. Bale, A. Mets, M. Verweij, D. Dorhout, J. Ossebaar, R. van

Bommel, Dr. M. van der Meer, I. Posthuma, A. Nordeloos, and M. Grego for technical support and advice.

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

        Availability on Methane-Influenced Microorganisms in an Enrichment Culture Obtained From a Stratified

Lake, Frontiers in Microbiology, 11, 2020.

Guggenheim, C., Freimann, R., Mayr, M. J., Beck, K., Wehrli, B., and Bürgmann, H.: Environmental and
        Microbial Interactions Shape Methane-Oxidizing Bacterial Communities in a Stratified Lake, Frontiers in
        Microbiology, 11, 2020.

Hanson, R. S.: Ecology and Diversity of Methylotrophic Organisms, in: Advances in Applied Microbiology,

vol. 26, edited by: Perlman, D., Academic Press, 3–39, https://doi.org/10.1016/S0065-2164(08)70328-9,
        1980.

Hanson, R. S. and Hanson, T. E.: Methanotrophic bacteria, Microbiological Reviews, 60, 439–471,
        https://doi.org/10.1128/mr.60.2.439-471.1996, 1996.

Harrits, S. M. and Hanson, R. S.: Stratification of aerobic methane-oxidizing organisms in Lake Mendota,

Madison, Wisconsin1, Limnology and Oceanography, 25, 412–421,
        https://doi.org/10.4319/lo.1980.25.3.0412, 1980.



Hernández, A., Kutiel, H., Trigo, R. M., Valente, M. A., Sigró, J., Cropper, T., and Santo, F. E.: New Azores archipelago daily precipitation dataset and its links with large-scale modes of climate variability, International Journal of Climatology, 36, 4439–4454, https://doi.org/10.1002/joc.4642, 2016.

Hopmans, E. C., Smit, N. T., Schwartz-Narbonne, R., Sinninghe Damsté, J. S., and Rush, D.: Analysis of non-derivatized bacteriohopanepolyols using UHPLC-HRMS reveals great structural diversity in environmental lipid assemblages, Organic Geochemistry, 160, 104285, https://doi.org/10.1016/j.orggeochem.2021.104285, 2021.

Kusch, S. and Rush, D.: Revisiting the precursors of the most abundant natural products on Earth: A look back

at 30+ years of bacteriohopanepolyol (BHP) research and ahead to new frontiers, Organic Geochemistry, 172, 104469, https://doi.org/10.1016/j.orggeochem.2022.104469, 2022.

Kusch, S., Sepúlveda, J., and Wakeham, S. G.: Origin of Sedimentary BHPs Along a Mississippi River–Gulf of Mexico Export Transect: Insights From Spatial and Density Distributions, Frontiers in Marine Science, 6, 2019.

Kusch, S., Wakeham, S. G., Dildar, N., Zhu, C., and Sepúlveda, J.: Bacterial and archaeal lipids trace chemo(auto)trophy along the redoxcline in Vancouver Island fjords, Geobiology, 19, 521–541, https://doi.org/10.1111/gbi.12446, 2021a.

Kusch, S., Wakeham, S. G., and Sepúlveda, J.: Diverse origins of "soil marker" bacteriohopanepolyols in marine oxygen deficient zones, Organic Geochemistry, 151, 104150,

https://doi.org/10.1016/j.orggeochem.2020.104150, 2021b.

Matys, E. D., Sepúlveda, J., Pantoja, S., Lange, C. B., Caniupán, M., Lamy, F., and Summons, R. E.: Bacteriohopanepolyols along redox gradients in the Humboldt Current System off northern Chile, Geobiology, 15, 844–857, https://doi.org/10.1111/gbi.12250, 2017.

Meyers, P. A.: Applications of organic geochemistry to paleolimnological reconstructions: a summary of

examples from the Laurentian Great Lakes, Organic Geochemistry, 34, 261–289, https://doi.org/10.1016/S0146-6380(02)00168-7, 2003.

Mitrović, D., Hopmans, E.C., Bale, N.J., Richter, N., Amaral-Zettler, L.A., Baxter, A.J., Peterse, F., Raposeiro, P.M., Gonçalves, V., Costa, A.C., and Schouten, S.: Isoprenoidal GDGTs and GDDs associated with anoxic lacustrine environments, *in review*.

National Center for Biotechnology Information. PubChem Compound Summary for CID 641496, 1,3-Oxazinan-2-one: https://pubchem.ncbi.nlm.nih.gov/compound/1_3-Oxazinan-2-one, last access: 26 January 2023a.

National Center for Biotechnology Information. PubChem Compound Summary for CID 123834, 1,3-Dioxan-2-one: https://pubchem.ncbi.nlm.nih.gov/compound/1_3-Dioxan-2-one, last access: 26 January 2023b.

Neunlist, S. and Rohmer, M.: A novel hopanoid, 30-(5'-adenosyl)hopane, from the purple non-sulphur

bacterium *Rhodopseudomonas acidophila*, with possible DNA interactions., Biochem J, 228, 769–771, 1985a.

Neunlist, S. and Rohmer, M.: Novel hopanoids from the methylotrophic bacteria *Methylococcus capsulatus* and *Methylomonas methanica*. (22S)-35-aminobacteriohopane-30,31,32,33,34-pentol and (22S)-35-amino-3β-methylbacteriohopane-30,31,32,33,34-pentol, Biochemical Journal, 231, 635–639,

https://doi.org/10.1042/bj2310635, 1985b.





Neunlist, S. and Rohmer, M.: The Hopanoids of '*Methylosinus trichosporium*': Aminobacteriohopanetriol and
Aminobacteriohopanetetrol, Microbiology, 131, 1363–1367, https://doi.org/10.1099/00221287-131-6-1363,
1985c.

Neunlist, S., Bisseret, P., and Rohmer, M.: The hopanoids of the purple non-sulfur bacteria *Rhodopseudomonas*
*palustris* and *Rhodopseudomonas acidophila* and the absolute configuration of bacteriohopanetetrol,
European Journal of Biochemistry, 171, 245–252, https://doi.org/10.1111/j.1432-1033.1988.tb13783.x,
1988.

O'Beirne, M. D., Sparkes, R., Hamilton, T. L., van Dongen, B. E., Gilhooly, W. P., and Werne, J. P.:
Characterization of diverse bacteriohopanepolyols in a permanently stratified, hyper-euxinic lake, Organic
Geochemistry, 168, 104431, https://doi.org/10.1016/j.orggeochem.2022.104431, 2022.

Osborne, K. A.: Environmental controls on bacteriohopanepolyol signatures in estuarine sediments, Newcastle
University, UK, 2015.

Osborne, K. A., Gray, N. D., Sherry, A., Leary, P., Mejeha, O., Bischoff, J., Rush, D., Sidgwick, F. R., Birgel,
D., Kalyuzhnaya, M. G., and Talbot, H. M.: Methanotroph-derived bacteriohopanepolyol signatures as a
function of temperature related growth, survival, cell death and preservation in the geological record,
Environmental Microbiology Reports, 9, 492–500, https://doi.org/10.1111/1758-2229.12570, 2017.

Oswald, K., Milucka, J., Brand, A., Littmann, S., Wehrli, B., Kuypers, M. M. M., and Schubert, C. J.: Light-
Dependent Aerobic Methane Oxidation Reduces Methane Emissions from Seasonally Stratified Lakes,
PLOS ONE, 10, e0132574, https://doi.org/10.1371/journal.pone.0132574, 2015.

Oswald, K., Milucka, J., Brand, A., Hach, P., Littmann, S., Wehrli, B., Kuypers, M. M. M., and Schubert, C. J.:
Aerobic gammaproteobacterial methanotrophs mitigate methane emissions from oxic and anoxic lake
waters, Limnology and Oceanography, 61, S101–S118, https://doi.org/10.1002/lno.10312, 2016.

Ourisson, G. and Albrecht, P.: Hopanoids. 1. Geohopanoids: the most abundant natural products on Earth?, Acc.
Chem. Res., 25, 398–402, https://doi.org/10.1021/ar00021a003, 1992.

Pearson, A., Leavitt, W. D., Sáenz, J. P., Summons, R. E., Tam, M. C.-M., and Close, H. G.: Diversity of
hopanoids and squalene-hopene cyclases across a tropical land-sea gradient, Environmental Microbiology,
11, 1208–1223, https://doi.org/10.1111/j.1462-2920.2008.01817.x, 2009.

Pereira, C. L., Raposeiro, P. M., Costa, A. C., Bao, R., Giralt, S., and Gonçalves, V.: Biogeography and lake
morphometry drive diatom and chironomid assemblages' composition in lacustrine surface sediments of
oceanic islands, Hydrobiologia, 730, 93–112, https://doi.org/10.1007/s10750-014-1824-6, 2014.

Raposeiro, P. M., Rubio, M. J., González, A., Hernández, A., Sánchez-López, G., Vázquez-Loureiro, D., Rull,
V., Bao, R., Costa, A. C., Gonçalves, V., Sáez, A., and Giralt, S.: Impact of the historical introduction of
exotic fishes on the chironomid community of Lake Azul (Azores Islands), Palaeogeography,
Palaeoclimatology, Palaeoecology, 466, 77–88, https://doi.org/10.1016/j.palaeo.2016.11.015, 2017.

Raposeiro, P. M., Saez, A., Giralt, S., Costa, A. C., and Gonçalves, V.: Causes of spatial distribution of subfossil
diatom and chironomid assemblages in surface sediments of a remote deep island lake, Hydrobiologia, 815,
141–163, https://doi.org/10.1007/s10750-018-3557-4, 2018.

Rethemeyer, J., Schubotz, F., Talbot, H. M., Cooke, M. P., Hinrichs, K.-U., and Mollenhauer, G.: Distribution
of polar membrane lipids in permafrost soils and sediments of a small high Arctic catchment, Organic
Geochemistry, 41, 1130–1145, https://doi.org/10.1016/j.orggeochem.2010.06.004, 2010.





Richter, N., Russell, J. M., Amaral-Zettler, L., DeGroff, W., Raposeiro, P. M., Gonçalves, V., de Boer, E. J., Pla-Rabes, S., Hernández, A., Benavente, M., Ritter, C., Sáez, A., Bao, R., Trigo, R. M., Prego, R., and Giralt, S.: Long-term hydroclimate variability in the sub-tropical North Atlantic and anthropogenic impacts on lake ecosystems: A case study from Flores Island, the Azores, Quaternary Science Reviews, 285, 107525,
https://doi.org/10.1016/j.quascirev.2022.107525, 2022.

Rohmer, M., Bouvier-Nave, P., and Ourisson, G. 1984: Distribution of Hopanoid Triterpenes in Prokaryotes, Microbiology, 130, 1137–1150, https://doi.org/10.1099/00221287-130-5-1137, 1984.

Rudd, J. W. M., Furutani, A., Flett, R. J., and Hamilton, R. D.: Factors controlling methane oxidation in shield lakes: The role of nitrogen fixation and oxygen concentration1, Limnology and Oceanography, 21, 357–364,
https://doi.org/10.4319/lo.1976.21.3.0357, 1976.

Rush, D., Osborne, K. A., Birgel, D., Kappler, A., Hirayama, H., Peckmann, J., Poulton, S. W., Nickel, J. C., Mangelsdorf, K., Kalyuzhnaya, M., Sidgwick, F. R., and Talbot, H. M.: The Bacteriohopanepolyol Inventory of Novel Aerobic Methane Oxidising Bacteria Reveals New Biomarker Signatures of Aerobic Methanotrophy in Marine Systems, PLOS ONE, 11, e0165635,
https://doi.org/10.1371/journal.pone.0165635, 2016.

Rush, D., Talbot, H. M., van der Meer, M. T. J., Hopmans, E. C., Douglas, B., and Sinninghe Damsté, J. S.: Biomarker evidence for the occurrence of anaerobic ammonium oxidation in the eastern Mediterranean Sea during Quaternary and Pliocene sapropel formation, Biogeosciences, 16, 2467–2479, https://doi.org/10.5194/bg-16-2467-2019, 2019.

Sáenz, J. P.: Hopanoid enrichment in a detergent resistant membrane fraction of *Crocosphaera watsonii*: Implications for bacterial lipid raft formation, Organic Geochemistry, 41, 853–856, https://doi.org/10.1016/j.orggeochem.2010.05.005, 2010.

Sáenz, J. P., Eglinton, T. I., and Summons, R. E.: Abundance and structural diversity of bacteriohopanepolyols in suspended particulate matter along a river to ocean transect, Organic Geochemistry, 42, 774–780,
https://doi.org/10.1016/j.orggeochem.2011.05.006, 2011.

Sáenz, J. P., Sezgin, E., Schwille, P., and Simons, K.: Functional convergence of hopanoids and sterols in membrane ordering, Proceedings of the National Academy of Sciences, 109, 14236–14240, https://doi.org/10.1073/pnas.1212141109, 2012.

Santos, F. D., Valente, M. A., Miranda, P. M. A., Aguiar, A., Azevedo, E. B., Tomé, A. R., and Coelho, F.:
Climate change scenarios in the Azores and Madeira Islands, World Resources Review, 16, 19, 2004.

Seemann, M., Bisseret, P., Tritz, J.-P., Hooper, A. B., and Rohmer, M.: Novel bacterial triterpenoids of the hopane series from *Nitrosomonas europaea* and their significance for the formation of the C35 bacteriohopane skeleton, Tetrahedron Letters, 40, 1681–1684, https://doi.org/10.1016/S0040-4039(99)00064-7, 1999.

Sinninghe Damsté, J. S., Rijpstra, W. I. C., Dedysh, S. N., Foesel, B. U., and Villanueva, L.: Pheno- and Genotyping of Hopanoid Production in Acidobacteria, Frontiers in Microbiology, 8, 2017.

Spencer-Jones, C. L., Wagner, T., Dinga, B. J., Schefuß, E., Mann, P. J., Poulsen, J. R., Spencer, R. G. M., Wabakanghanzi, J. N., and Talbot, H. M.: Bacteriohopanepolyols in tropical soils and sediments from the Congo River catchment area, Organic Geochemistry, 89–90, 1–13,
https://doi.org/10.1016/j.orggeochem.2015.09.003, 2015.

Summons, R. E., Jahnke, L. L., Hope, J. M., and Logan, G. A.: 2-Methylhopanoids as biomarkers for cyanobacterial oxygenic photosynthesis, Nature, 400, 554–557, https://doi.org/10.1038/23005, 1999.

Talbot, H. M. and Farrimond, P.: Bacterial populations recorded in diverse sedimentary biohopanoid distributions, Organic Geochemistry, 38, 1212–1225, https://doi.org/10.1016/j.orggeochem.2007.04.006, 2007.

Talbot, H. M., Watson, D. F., Murrell, J. C., Carter, J. F., and Farrimond, P.: Analysis of intact bacteriohopanepolyols from methanotrophic bacteria by reversed-phase high-performance liquid chromatography–atmospheric pressure chemical ionisation mass spectrometry, Journal of Chromatography A, 921, 175–185, https://doi.org/10.1016/S0021-9673(01)00871-8, 2001.

Talbot, H. M., Watson, D. F., Pearson, E. J., and Farrimond, P.: Diverse biohopanoid compositions of non-marine sediments, Organic Geochemistry, 34, 1353–1371, https://doi.org/10.1016/S0146-6380(03)00159-1, 2003.

Talbot, H. M., Rohmer, M., and Farrimond, P.: Rapid structural elucidation of composite bacterial hopanoids by atmospheric pressure chemical ionisation liquid chromatography/ion trap mass spectrometry, Rapid Communications in Mass Spectrometry, 21, 880–892, https://doi.org/10.1002/rcm.2911, 2007.

Talbot, H. M., Summons, R. E., Jahnke, L. L., Cockell, C. S., Rohmer, M., and Farrimond, P.: Cyanobacterial bacteriohopanepolyol signatures from cultures and natural environmental settings, Organic Geochemistry, 39, 232–263, https://doi.org/10.1016/j.orggeochem.2007.08.006, 2008.

Talbot, H. M., Handley, L., Spencer-Jones, C. L., Dinga, B. J., Schefuß, E., Mann, P. J., Poulsen, J. R., Spencer, R. G. M., Wabakanghanzi, J. N., and Wagner, T.: Variability in aerobic methane oxidation over the past 1.2Myrs recorded in microbial biomarker signatures from Congo fan sediments, Geochimica et Cosmochimica Acta, 133, 387–401, https://doi.org/10.1016/j.gca.2014.02.035, 2014.

Taylor, K. A. and Harvey, H. R.: Bacterial hopanoids as tracers of organic carbon sources and processing across the western Arctic continental shelf, Organic Geochemistry, 42, 487–497, https://doi.org/10.1016/j.orggeochem.2011.03.012, 2011.

Wagner, T., Kallweit, W., Talbot, H. M., Mollenhauer, G., Boom, A., and Zabel, M.: Microbial biomarkers support organic carbon transport from methane-rich Amazon wetlands to the shelf and deep sea fan during recent and glacial climate conditions, Organic Geochemistry, 67, 85–98, https://doi.org/10.1016/j.orggeochem.2013.12.003, 2014.

Watson, D. F. and Farrimond, P.: Novel polyfunctionalised geohopanoids in a recent lacustrine sediment (Priest Pot, UK), Organic Geochemistry, 31, 1247–1252, https://doi.org/10.1016/S0146-6380(00)00148-0, 2000.

Welander, P. V. and Summons, R. E.: Discovery, taxonomic distribution, and phenotypic characterization of a gene required for 3-methylhopanoid production, Proceedings of the National Academy of Sciences, 109, 12905–12910, https://doi.org/10.1073/pnas.1208255109, 2012.

Welander, P. V., Hunter, R. C., Zhang, L., Sessions, A. L., Summons, R. E., and Newman, D. K.: Hopanoids Play a Role in Membrane Integrity and pH Homeostasis in *Rhodopseudomonas palustris* TIE-1, Journal of Bacteriology, 191, 6145–6156, https://doi.org/10.1128/JB.00460-09, 2009.

Welander, P. V., Coleman, M. L., Sessions, A. L., Summons, R. E., and Newman, D. K.: Identification of a methylase required for 2-methylhopanoid production and implications for the interpretation of sedimentary hopanes, Proc. Natl. Acad. Sci. U.S.A., 107, 8537–8542, https://doi.org/10.1073/pnas.0912949107, 2010.



Whittenbury, R., Phillips, K. C., and Wilkinson, J. F.: Enrichment, Isolation and Some Properties of Methane-utilizing Bacteria, Journal of General Microbiology, 61, 205–218, https://doi.org/10.1099/00221287-61-2-205, 1970.

van Winden, J. F., Talbot, H. M., Kip, N., Reichart, G.-J., Pol, A., McNamara, N. P., Jetten, M. S. M., Op den Camp, H. J. M., and Sinninghe Damsté, J. S.: Bacteriohopanepolyol signatures as markers for methanotrophic bacteria in peat moss, Geochimica et Cosmochimica Acta, 77, 52–61, https://doi.org/10.1016/j.gca.2011.10.026, 2012.

van Winden, J. F., Talbot, H. M., Reichart, G.-J., McNamara, N. P., Benthien, A., and Sinninghe Damsté, J. S.: Influence of temperature on the $\delta^{13}C$ values and distribution of methanotroph-related hopanoids in Sphagnum-dominated peat bogs, Geobiology, 18, 497–507, https://doi.org/10.1111/gbi.12389, 2020.

Xu, Y., Cooke, M. P., Talbot, H. M., and Simpson, M. J.: Bacteriohopanepolyol signatures of bacterial populations in Western Canadian soils, Organic Geochemistry, 40, 79–86, https://doi.org/10.1016/j.orggeochem.2008.09.003, 2009.

Zhu, C., Talbot, H. M., Wagner, T., Pan, J.-M., and Pancost, R. D.: Distribution of hopanoids along a land to sea transect: Implications for microbial ecology and the use of hopanoids in environmental studies, Limnology and Oceanography, 56, 1850–1865, https://doi.org/10.4319/lo.2011.56.5.1850, 2011.

Zindorf, M., Rush, D., Jaeger, J., Mix, A., Penkrot, M. L., Schnetger, B., Sidgwick, F. R., Talbot, H. M., van der Land, C., Wagner, T., Walczak, M., and März, C.: Reconstructing oxygen deficiency in the glacial Gulf of Alaska: Combining biomarkers and trace metals as paleo-redox proxies, Chemical Geology, 558, 119864, https://doi.org/10.1016/j.chemgeo.2020.119864, 2020.

Zundel, M. and Rohmer, M.: Prokaryotic triterpenoids, European Journal of Biochemistry, 150, 23–27, https://doi.org/10.1111/j.1432-1033.1985.tb08980.x, 1985.