# Peer review of "Distributions of bacteriohopanepolyols in lakes and coastal lagoons of the Azores Archipelago"

_Biogeosciences, 2023_

## Author Response (AR1)

Dear Dr Richter et al.,

Thank you for submitting your manuscript to Biogeosciences and for responding to the comments/suggestions from both reviewers.

Both reviewers recommended minor revisions, which is also in agreement with my personal review. My questions and comments have already been covered by the reviewers. I think you have appropriately addressed the reviewer's feedback and I look forward to the revised manuscript which includes the changes as per your responses.

Thank you for these edits, and for your support of Biogeosciences.

Sincerely,

Dr Sebastian Naeher
Associate Editor, Biogeosciences

*We would like to thank both the editor and the reviewers for their feedback on our manuscript. We have made the modifications as outlined below and in the revised manuscript.*

**Reviewer #1**

**General comments**

The study by Richter et al. describes distributions of BHPs in lacustrine and coastal environments of the Azores. In addition to the environmental samples, an enrichment culture enriched for methanotrophs was also studied. Using LCMS, the authors identified several novel BHPs and report detailed information of their mass spectral identification. The BHP distributions are then discussed in combination with geochemical parameters and the potential as taxonomic markers for different ecological niches is evaluated.

I like the manuscript and think it is well written and structured. The type of figures and the selected statistical methods are chosen well. The detailed identification of the new compounds seems sound and the reported mass spectra in the appendix will be very useful for further research. I am very much looking forward to the final version where the raw data are also accessible. Besides the analytical part, the discussion of the findings in context of previous studies is well balanced and adds important conclusions. Overall, not much should be changed before final publication.

We would like to the thank the reviewer for taking the time to review our manuscript and for their thoughtful and positive comments.

**Specific comments**

It would be great if the authors could also make the inclusion lists for the analytical method available (see also minor comment). This would help "jump-start" implementation of the LCMS method in other labs.

The appendix is great in explaining in detail how the fragment spectra of the BHPs are used for identification. The main text then only contains a brief mentioning (according to line 212) – this is true for all novel compounds except the formylated-aminoBHPs where lines 262-278 are very similar to section B6 in the appendix. I suggest to shorten the main text version so that it is similar to the summary of findings of the other compounds and keep the details in the appendix B6.

Thank you for the recommendation, we will modify the main text and the appendix. We expanded on the formylated-aminoBHPs in the main text because this is the first time this class of compounds is described. Versions of the other novel compounds discussed in this manuscript were previously described by Hopmans et al. (2021) and Elling et al. (2022). Therefore, we would like to keep some of the more expanded description in the main text of the manuscript, but we will shorten it to make it easier to follow. We will also make the inclusion list available as part of the supplementary materials for this manuscript.

***We modified the main text as follows:***

"In sediment from Lake Verde (Fig. 3, peaks j and m), we identified several unknown composite BHPs previously described in Hopmans et al. (2021) with an assigned elemental composition (AEC) of $C_{36}H_{64}O_4N^+$ (*m/z* 574.483) and $C_{36}H_{64}O_5N^+$ (*m/z* 590.483), but no structure was assigned. In addition, we identified the same compound but with an additional hydroxy moiety (Fig. 3, peak o). We propose that these novel composite BHPs are a series of N-formylated-aminoBHPs: N-formylated-aminotriol (peak j), N-formylated-aminotetrol (peak m) and N-formylated-aminopentol (peak o). The proposed structure for N-formylated-aminopentol is shown in Fig. 3f with the diagnostic fragmentations indicated. In addition, we observe unsaturated versions of N-formylated-aminotriol (Fig. 3b & Appendix B5, Fig. B7), N-formylated-aminotetrol (based on the $MS^1$ and retention time), and N-formylated-aminopentol with the double bond likely occurring at the $\Delta^6$ position for N-formylated-aminotriol and $\Delta^{11}$ position for N-formylated-aminotetrol and N-formylated-aminopentol. In the mass chromatogram for *m/z* 572.467 (Fig. 3b) from the Lake Verde sediment, we also observe a later-eluting peak ($C_{36}H_{62}O_4N^+$; Fig. 3b and d, peak l). Based on the $MS^2$ spectrum, we propose that this is the same compound identified by Elling et al. (2022) in nitrite-oxidizing bacteria. This compound is likely a cyclized form of N-formylated-aminotriol, and we putatively identify it as oxazinone-aminotriol (Appendix B6).

In addition to the discussed $\Delta^6$-ethenolamine-BHT, we observed a later eluting peak at 21.88 mins (Fig. 3a, peak c) in the mass chromatogram of *m/z* 586.483 ($C_{37}H_{64}O_4N^+$; $\Delta$ ppm 0.72). We propose that one of the functionalities is part of a cyclized structure and this BHP is a N-formylated aminotetrol, where the formic acid is part of a cyclic structure and the terminal amino group is methylated. The proposed structure with key fragmentations is shown in Fig. 3c. We tentatively identify this compound as a dioxanone-methylaminotriol (Appendix B7)." (lines 273-293)

***We have added the inclusion list to the supplementary material.***

Since there are so many compounds in the NMDS plot of Fig. 4, I wonder if it is possible to use colored font for the compounds clustering together in Fig. 4. For example color A for cluster of amino-containing compounds (referenced in line 323), color B for the nucleoside

compounds (line 338) and so on (e.g. in line 466). This should then be indicated in the text (section lines 314-349) and in the caption of Fig. 4.

Thank you for the suggestion, we will modify the figure as recommended by the reviewer. We agree that this will make the figure easier to interpret.

*We have modified the colors for the different compound classes as recommended by the reviewer.*

Section 3.5 (implications for the Rsoil proxy) is not reflected in the abstract and conclusions and should be added.

Thank you for the suggestion. We will add a sentence about the Rsoil proxy to the abstract and conclusions.

*We have added the following sentences to the abstract and conclusions:*

*Abstract:*

"Based on our current results we revised the existing $R_{soil}$ index, which was previously used to infer terrestrial inputs to aquatic environments, to exclude any potential nucleosides produced in the lake water column ($R_{soil-lake}$)." (lines 24-26)

*Conclusions:*

"Further, we identified several nucleoside BHPs that appear to be produced in the water column of lacustrine settings, however, further work is needed to verify the source of these BHPs. We propose a revised $R_{soil}$ index, $R_{soil-lake}$, to distinguish between terrestrial- and aquatic-derived organic matter in lakes, by excluding any nucleosides that might be produced *in situ*." (lines 618-621)

**Minor comments and technical corrections**

Thank you for the minor comments and technical corrections, we will address these in the revised manuscript.

L67: replace "nor " with "or"

We will correct this.

*Corrected*

L154: specify MS model?

We will add this.

*Added:* "quadrupole-orbitrap HRMS (Q-Exactive, ThermoFisher Scientific, Waltham, MA)" (line 161-162)

L159: resolution does not have unit "ppm"

The unit "ppm" is present, but was moved to the following line.

L159-160: why three separate inclusion lists? Can these lists be provided as supplemental material, other research would benefit a lot from this

The surface sediment samples were analyzed initially with the inclusion list we had available at the time. After the identification of several novel compounds, we analyzed the water column samples with an updated inclusion list. Finally, we re-analyzed a select number of samples to obtain a better MS2 and verify our tentative identification of these novel compounds. We have modified this in the text to clarify this point. We will include the final inclusion list in the supplementary material.

*We modified the text as follows to clarify the use of different inclusion lists:*

"Lipids were detected using positive ion monitoring of *m/z* 350-2000 (resolution 70,000 ppm at *m/z* 200) with an inclusion list of calculated exact masses of BHPs: 171 for sediment samples and 357 for water column samples. Note, that the difference in the number of exact masses in the inclusion lists reflects the addition of further novel BHPs after their identification. A Lake Verde sediment sample (Verde SS1) that included all the novel BHPs was re-run with an updated inclusion list (421 exact masses) for confirmation of novel BHPs." (lines 166-170)

*As previously mentioned, we included the inclusion list in the supplementary material.*

L169: how was the normalization between sample runs done? Please explain.

We will expand on this in the text. Briefly, we used the same internal standard for all of our sample runs. We used the internal standard to normalize the measured peak areas of our BHPs.

*We added the following sentence:*

"We used an internal standard for normalization between sample runs by calculating the average peak area of the internal standard and correcting the peak areas to the average internal standard." (lines 179-180)

L256: Fig 3e? I might have missed it, but Fig. 3e is not referenced?

Thank you for noticing this mistake, we will correct this in the manuscript.

*We added the figure reference to the following sentence:*

"In addition, we putatively identified unsaturated versions of ethenolamine-BHT (Appendix B4, Fig. B6) and ethenolamine-BHhexol (Fig. 3e) with the double bond likely occurring at the $\Delta^6$ and $\Delta^{11}$ positions, respectively, based on the retention times." (lines 272-276)

L303-305: "Using ANOSIM we find a significant difference…"

We will correct this.

*Corrected.*

Fig. 4: colors are not easy to distinguish (two shades of green, two of blue), choose additional other colors

Thank you for the suggestion, we will modify the colors in this figure.

*We modified the colors and shapes for the lakes.*

L311: delete "there is": "we find no significant difference"

Will correct.

*Corrected.*

L327: "was previously described"

Will correct.

*Corrected.*

Fig. 6 caption: RU should be "response units", not "relative units"

Will correct.

*Corrected.*

Fig. 9 caption: indicate that structures are tentative structures

Will correct.

*Corrected.*

L478: instead of "an NMDS" use "the NMDS analysis". This also reads a bit odd, the NMDS analysis shows many compounds. Do you mean that these compounds, the amino BHP cluster, is close to the sites mentioned in line 482?

Yes, what we meant to say is that ethenolamine-BHpentol, ethenolamine-BHhexol, acylated-ethenolamine BHhexols ($C_{15:0}$, $C_{16:0}$, $C_{17:0}$), formylated-aminotetrol, formylated-aminopentol, MC-aminopentol, aminopentol, aminotetrol and the acylated-aminopentols all cluster near the surface sediments from Azul, Verde, Funda, and Negra and the bottom water from Funda and Negra. We will rephrase this in the text to make it clearer.

*We modified the text as follows:*

"The widespread distribution of these compounds suggests they are derived from multiple producers. Although we cannot identify a direct source for these compounds, the NMDS analysis shows that ethenolamine-BHpentol, ethenolamine-BHhexol, acylated-ethenolamine BHhexols ($C_{15:0}$, $C_{16:0}$, $C_{17:0}$), N-formylated-aminotetrol, and N-formylated-aminopentol cluster with MC-aminopentol, aminopentol, aminotetrol and the acylated-aminopentols near

the surface sediments from Azul, Verde, Funda, and Negra (Fig. 4). In addition, ethenolamine-BHhexol and N-formylated-aminopentol are detected in the bottom waters of Lake Funda (Fig. 6). Azul, Verde, Funda, and Negra all stratify during the summer months and the bottom water in these lakes range from suboxic to anoxic conditions (Gonçalves et al., 2018). This would suggest that ethenolamine-BHpentol, ethenolamine-BHhexol, acylated-ethenolamine BHhexols, N-formylated-aminotetrol, and N-formylated-aminopentol are being produced under suboxic to anoxic conditions and/or could be associated with MOB." (lines 492-502)

L493: suggestion – "observe … in all sediment samples except Cubres East and West (Fig. 2)…", move outside of parentheses

Will correct.

*Corrected.*

L538: "BHP distribution"

Will correct.

*Corrected.*

L603: perhaps "… appear to be produced in the water column…"

Will correct.

*Corrected.*

Appendix B1 second paragraph last sentence: typo – "…attribute this to a potential co-eluting…are unable to distinguish."

Will correct.

*Corrected.*

Appendix B114: m/z 191 was searched in fragment spectra: "…in fragment spectra revealed two…"

Will correct.

*Corrected.*

Appendix B120: "..is not located in the ring system."

Will correct.

*Corrected.*

**References**

Elling, F. J., Evans, T. W., Nathan, V., Hemingway, J. D., Kharbush, J. J., Bayer, B., Spieck, E., Husain, F., Summons, R. E., and Pearson, A.: Marine and terrestrial nitrifying bacteria are sources of diverse bacteriohopanepolyols, Geobiology, 20, 399–420, https://doi.org/10.1111/gbi.12484, 2022.

Hopmans, E. C., Smit, N. T., Schwartz-Narbonne, R., Sinninghe Damsté, J. S., and Rush, D.: Analysis of non-derivatized bacteriohopanepolyols using UHPLC-HRMS reveals great structural diversity in environmental lipid assemblages, Organic Geochemistry, 160, 104285, https://doi.org/10.1016/j.orggeochem.2021.104285, 2021.

**Reviewer #2**

In an effort to identify BHPs that can be associated with specific ecological niches or bacteria, Richter and colleagues have characterized the BHP distributions in suspended particle matter and sediments in the lakes and lagoons of the Azores Archipelago. The authors are particularly interested in characterizing BHP distributions and their potential as taxonomic markers in lacustrine settings, which have not been as well studied as marine settings. This study identifies multiple novel BHPs, including ethenolamine-BHPs and formylated-aminoBHPs, and proposes potential taxonomic markers for methane-oxidizing bacteria and low-oxygen conditions in lacustrine settings. In addition, they identify nucleoside-BHPs that can be produced within the water column, which has implications for the $R_{soil}$ proxy, leading the authors to propose a new $R_{soil-lake}$ proxy. The results of this study will be useful for future work using BHPs to characterize lacustrine settings and I recommend it be published in *Biogeosciences* with minor revisions. My general recommendation is to consider what additional information may be available, like $\delta^{13}C_{TOC}$ or metagenomic data, that could be used to support the potential of certain BHPs as proxies, especially for MOBs.

We would like to thank the reviewer for taking the time to review our manuscript and their constructive comments.

Specific Comments:

Line 61: Do both type I and type II MOB produce the same diagnostic BHPs or do we know of differences between them? If there are known differences, be more specific when discussing the diagnostic BHPs. Lines 421-422 suggest that they produce different diagnostic BHPs, but this should be consistently clear.

Thank you for pointing this out. As mentioned in lines 421-422, it was previously proposed that type I and type II MOB produce diagnostic BHPs, i.e. aminopentol and aminotetrol, respectively. However, more recent studies (see Rush et al., 2016; Kusch and Rush, 2022), show that this previous distinction is not as clear as previously thought. More work is therefore needed to test whether other proxies can provide us with additional information on being able to distinguish between type I and type II MOB. As suggested by the reviewer we will modify the text to clarify this point.

*We modified the text as follows:*

"Type I methanotrophs are typically associated with increased production of aminopentol, whereas Type II methanotrophs are distinguished by the production of aminotetrol (Neunlist and Rohmer, 1985b, c; Cvejic et al., 2000; Talbot et al., 2001; van Winden et al., 2012). However, recent environmental studies, particularly in marine environments, suggest that this distinction is not as clear as previously proposed (Rush et al., 2016; Kusch and Rush, 2022)." (lines 67-71)

Line 65: The phrase "minor amounts of these BHPs" is vague. Are all of the BHPs mentioned in the previous sentence (aminotetrol, aminopentol, methylcarbamate-aminoBHPs, and 3β-methylated-BHPs) also produced by sulfur-reducing bacteria and nitrite-oxidizing bacteria or just a subset of them? This should be clarified.

Only aminotetrol and aminopentol are produced by sulfur-reducing bacteria in low amounts (Blumenberg et al., 2006). Methylcarbamate-aminotriol was identified in nitrate-oxidizing bacteria by Elling et al. (2022). We will modify the text in the manuscript to clarify this point.

*We modified the text as follows:*

"In addition, minor amounts aminotetrol and aminopentol are also produced by sulfur-reducing bacteria and methylcarbamate-aminotriol was found in cultures of nitrite-oxidizing bacteria (Blumenberg et al., 2006; Elling et al., 2022)." (lines 71-74)

Lines 113-114: It is mentioned that Lakes Verde and Funda experience large cyanobacterial blooms in the summer months. Were either of these lakes experiencing cyanobacterial blooms during sampling?

Yes, Lakes Verde and Funda were both experiencing cyanobacterial blooms during sampling. We will clarify this point in the text.

*We added the following sentence:*

"Both lakes Verde and Funda were experiencing a cyanobacterial bloom at the time of sampling." (lines 120-121)

Line 129: What do the given percentages mean for this co-culture? Were there other organisms present and this was the percentage of bacteria that could be identified? If so, could other bacteria present be producing BHPs in the culture?

This is an enrichment co-culture, so it does contain other bacteria. However, based on the previous study conducted by van Grinsven et al. (2020), *Methylobacter* sp. and *Methylotenera* sp. are the primary bacteria in this enrichment. It is possible that other bacteria are producing some of these BHPs; however, this is likely a minor contribution. Further, the primary BHP profile (i.e., high proportions of aminotriol and aminopentol) of *Methylobacter* sp. resembles that of pure MOB Type I cultures previously analyzed. Future studies, however, should analyze BHPs from pure cultures to verify the BHP profile that we describe in this study for *Methylobacter*. We will add a few sentences to the text to acknowledge this potential discrepancy.

*We added the following sentence:*

"We acknowledge that other bacteria were also present in the enrichment culture (see van Grinsven et al., 2020); however, the BHP contribution is likely minor compared to the more abundant *Methylobacter* sp. present." (lines 420-422)

Line 179: TOC content was measured, but no associated $\delta^{13}C_{TOC}$ measurements are reported. This could be a useful additional piece of information to aid in characterizing the MOB input in this environment.

$\delta^{13}C_{TOC}$ measurements are available, and we will add these to the manuscript to support our preliminary interpretations.

*We added the following sentences to the methods and the main text:*

"δ$^{13}$C and δ$^{15}$N values of bulk organic matter were measured on decalcified samples of surface sediments (n=18) using an Elementar Analyser (Elementar Vario Isotope Cube) coupled to an IRMS (Elementar Isoprime vision). Stable isotope ratios of $^{13}$C and $^{15}$N are expressed using the δ notation in units per mil, reported relative to Vienna Pee Dee belemnite (V-PDB) and atmospheric N$_2$, respectively. The results were normalized to certified standards and standard deviations were <0.05 ‰." (lines 192-196)

"The low sediment TOC/TN values (7 to 11) and isotopically depleted δ$^{13}$C values (-31 to -24 ‰; Fig. C2a), indicate predominantly *in situ* derived organic matter which is in accordance with the high levels of primary productivity observed in all of the lakes (Cordeiro et al., 2020; Meyers, 2003). Similarly, low δ$^{15}$N values (-1 to 2 ‰; Fig. C2b) are commonly associated with high-levels of nitrogen-fixation in accordance with the cyanobacterial blooms observed in the lakes during the summer months (Cordeiro et al., 2020; Meyers, 2003). The slightly more enriched δ$^{13}$C values (-19 to -15 ‰) in Cubres East and West are typical of marine algae, also suggesting that the organic matter is mainly derived from *in situ* production (Meyers, 1994)." (lines 592-597)

*We also added Fig. C2 to Appendix C.*

[Figure]

**Figure C2: Distribution of (A) δ$^{13}$C (‰, VPDB) and TOC/TN values and (B) δ$^{15}$N (‰, Air) and δ$^{13}$C (‰, VPDB) values for the surface sediment samples analysed in this study.**

Line 252: Are the acylated versions of aminopentol also commonly associated with methanotrophic activity?

Prior to this study, acylated-aminopentols have only been reported by Hopmans et al. (2021) from soil samples collected near a terrestrial methane seep. Based on this previous study and our current study, we hypothesize that acylated-aminopentols are likely associated with methanotrophic bacteria. However, we recommend that further studies are conducted on both environmental samples and cultures of methanotrophs to test whether acylated-aminopentols are actually associated with methanotrophic activity. We discuss this in detail in lines 472-479.

Lines 274-275: Is there any evidence apart from the putative oxazinone-aminotriol for nitrite-oxidizing bacteria in the sediments from Lake Verde? While Elling et al. (2022) identified the same compound in nitrite-oxidizing bacteria, the source of that BHP here is not necessarily

the same. Further comment on the potential source of oxazinone-aminotriol in this environment would be useful.

As far as we know, Elling et al. (2022) was the first study to report oxazinone-aminotriol in cultures and our study is the to report oxazinone-aminotriol in environmental samples. At this time, we have not done any additional molecular work on these samples and therefore it is difficult to speculate on other potential sources of oxazinone-aminotriol. We hope to explore this in future studies in more detail.

Line 314: The fieldwork was done at the beginning of the stratification season for deep lakes – is there evidence that the lakes had fully stratified at that point? If not, this could potentially explain the finding that there is no significant difference between the BHP distributions in the surface, chemocline, and bottom water.

The deep lakes (Azul, Funda, and Negra) were stratified at the time of sampling as demonstrated by the water column profiles shown in Fig. 6 and in Table A2 (we will include the full water column profile data as part of the supplementary material). However, as mentioned by the reviewer, it was early in the season and the bottom waters of both Funda and Azul, for instance were not anoxic. Our ANOSIM test is likely also not significant due to the limited number of samples analyzed in our study. Further work is needed to test whether the BHP distributions are significantly different between the surface water, chemocline, and bottom water in other lakes.

***We included the full water column profile data for Lakes Funda and Azul in the supplementary material (see Tables S5 and S6).***

Figure 4: It would be easier to distinguish between deep lakes, shallow lakes, and coastal lagoons if the sample site markers for type of site had a different shape.

Thank you for the suggestion, we will modify the figure as recommended by the reviewer.

***We have modified Fig. 4 as suggested by both reviewers 1 and 2.***

Line 353: Is it clear why methoxylated BHPs are only present in the lagoons?

At this point no, but it is interesting! Hopefully future studies in coastal settings and marine environments will help us understand if this is a feature unique to marine settings and/or if we can link this to a specific producer.

Lines 444, 452: It is noted that sulfur-reducing bacteria and nitrite-oxidizing bacteria could potentially contribute to the aminotriol, aminotetrol, and aminopentol in the water column, although it is unlikely. Is there any additional data available, like $\delta^{13}C_{TOC}$ measurements or metagenomic data, that could support the MOB origin for these BHPs, especially for the potential origin from nitrite-oxidizing bacteria? Confirmation that nitrite-oxidizing bacteria are not making a significant contribution of aminopentol would also support the interpretation that that there is a higher abundance of Type I than Type II MOB.

$\delta^{13}C_{TOC}$ measurements are available, and we will add these to the manuscript to support our preliminary interpretations as mentioned previously.

*We have modified this as discussed above.*

Metagenomic work is currently beyond the scope of this study and molecular data is currently not available for these lakes. This is, however, a direction we are interested in pursuing as part of future work.

Line 524: A reference is made to the chironomid community shifting as a result of lake eutrophication. However, in the context of this paper, this line seems unrelated and would require more information on why chironomids would be relevant for understanding MOB contributions to the ecosystem. I would suggest either removing this line or further explaining the connection between chironomids and MOB.

Thank you for this suggestion, we will modify this in the text.

*We have removed this from the text.*

Technical Corrections:

Figure 1C: Sao Jorge Lake is not labeled or obvious in this panel, despite being included in the figure caption. Additionally, in the enlarged map, it would be useful to center and zoom in further on the lagoons of interest because they are currently difficult to see.

Thank you for the suggestion, we will modify this figure.

*We have modified this as suggested by the reviewer.*

Figure 1D: Is the enlarged map different from the map of Flores island? The island map has more lakes than the enlarged panel, which makes it challenging to identify the lakes of interest in the larger geographic context.

Thank you for pointing this out, we will modify this figure.

*We have modified this and labelled the lakes to make it easier to identify.*

Table 1: Formatting issues make this confusing, especially for the pH, $NH_4$, and $NO_2$ columns.

We will modify this to make it easier to read.

*We have moved the $NH_4$, $NO_3$, and $NO_2$ data to the supplementary material (Table S7).*

Lines 364-366: The caption for Fig. 6 says that dissolved oxygen in plots D and I and pH in plots E and J are pink when they are in black.

Thank you for noticing this mistake. We will correct this in the figure caption.

*Corrected.*

Appendix B1, page 31 (no line numbers): General typing errors – the appendix could use a proof-reading. Examples: "is similar to **(the)** retention time difference" and "we attribute this **to potential a potential** co-eluting peak."

Thank you for catching these mistakes. We will correct the abovementioned mistakes, and will proofread the Appendix for typing errors.

*We corrected these mistakes and proofread the appendix for additional typos.*

Figure B2A: The bottom chromatogram (m/z 718.563) does not have any labeled peaks. Based on retention time, I would assume it to be peak o, as labeled in the same chromatogram in panel B. If so, this should be labeled.

We will correct this in the manuscript.

*Corrected.*

**References**

Blumenberg, M., Krüger, M., Nauhaus, K., Talbot, H. M., Oppermann, B. I., Seifert, R., Pape, T., and Michaelis, W.: Biosynthesis of hopanoids by sulfate-reducing bacteria (genus Desulfovibrio), Environmental Microbiology, 8, 1220–1227, https://doi.org/10.1111/j.1462-2920.2006.01014.x, 2006.

Elling, F. J., Evans, T. W., Nathan, V., Hemingway, J. D., Kharbush, J. J., Bayer, B., Spieck, E., Husain, F., Summons, R. E., and Pearson, A.: Marine and terrestrial nitrifying bacteria are sources of diverse bacteriohopanepolyols, Geobiology, 20, 399–420, https://doi.org/10.1111/gbi.12484, 2022.

van Grinsven, S., Sinninghe Damsté, J. S., Harrison, J., and Villanueva, L.: Impact of Electron Acceptor Availability on Methane-Influenced Microorganisms in an Enrichment Culture Obtained From a Stratified Lake, Frontiers in Microbiology, 11, 2020.

Hopmans, E. C., Smit, N. T., Schwartz-Narbonne, R., Sinninghe Damsté, J. S., and Rush, D.: Analysis of non-derivatized bacteriohopanepolyols using UHPLC-HRMS reveals great structural diversity in environmental lipid assemblages, Organic Geochemistry, 160, 104285, https://doi.org/10.1016/j.orggeochem.2021.104285, 2021.

Kusch, S. and Rush, D.: Revisiting the precursors of the most abundant natural products on Earth: A look back at 30+ years of bacteriohopanepolyol (BHP) research and ahead to new frontiers, Organic Geochemistry, 172, 104469, https://doi.org/10.1016/j.orggeochem.2022.104469, 2022.

Rush, D., Osborne, K. A., Birgel, D., Kappler, A., Hirayama, H., Peckmann, J., Poulton, S. W., Nickel, J. C., Mangelsdorf, K., Kalyuzhnaya, M., Sidgwick, F. R., and Talbot, H. M.: The Bacteriohopanepolyol Inventory of Novel Aerobic Methane Oxidising Bacteria Reveals New Biomarker Signatures of Aerobic Methanotrophy in Marine Systems, PLOS ONE, 11, e0165635, https://doi.org/10.1371/journal.pone.0165635, 2016.

**Additional corrections:**

*We corrected additional typos and/or mistakes that we noted in the manuscript, see marked up version for details (e.g., changing the legend in Fig. 1 from "unknown composite" to "oxazinone-aminotriol). Further, we re-named formylated-aminoBHPs to N-formylated-aminoBHPs to specify that the formylation occurs on the amino-moiety.*